

# The Frosty Frontier: Redefining the Tropopause as a transport barrier using the Relative Humidity over Ice

Philipp Reutter[1] and Peter Spichtinger[1]

[1]Institute for Atmospheric Physics, Johannes Gutenberg University Mainz, Mainz, Germany

**Correspondence:** Philipp Reutter (preutter@uni-mainz.de)

**Abstract.** The tropopause acts as transport barrier between the troposphere and stratosphere. While common definitions rely on quantities conserved under adiabatic changes, diabatic effects, resulting from radiation, cloud processes or turbulence are also decisive for the tropopause structure. Therefore, we propose a new definition based on the vertical gradient of the relative humidity with respect to ice (RHi). RHi is the key variable for ice cloud formation and incorporates both diabatic and adiabatic

processes. Based on high-resolution radio sonde data we can show that our definition reflects the nature of the tropopause as a transport barrier much better than conventional approaches. This is not only evident in individual vertical profiles, but also when looking at statistics of many profiles with a tropopause-relative height axis. Last but not least, the robust and simple calculation of our definition makes it an ideal tool for studies involving the tropopause.

## 1   Introduction

The Earth's atmosphere extends from the ground to several hundred kilometers above the surface. It is characterized by several layers that are distinguished by thermal, chemical, and dynamic properties. The transition from one layer to another is usually abrupt and is noticeable through strong gradients in the respective variables. The active weather layer that determines our lives is the troposphere and extends from the ground to an altitude of about 8 to 12 km in the extratropics, depending on the seasons and current weather situation. In extreme cases, there are also strong upward and downward deviations from this altitude range. The

layer above is the stratosphere, which is characterized by very dry conditions and a general increase in temperature with altitude. The troposphere and stratosphere are separated by the tropopause, which generally acts as a transport barrier (Gettelman et al., 2011). The transfer between these two spheres, the so-called stratosphere-troposphere exchange (STE), affects key properties of the different layers. For instance, transport of water vapor from the troposphere into the stratosphere affects radiative properties of the stratosphere (Krebsbach et al., 2006), while the transport in the opposite direction can enhance the tropospheric ozone

concentration (Stevenson et al., 2006). Thus, the determination of the transport barrier is of high relevance for our understanding of the upper troposphere / lower stratosphere (UTLS) region.

There are many different ways to define the tropopause, mostly based on threshold values or gradient changes of different variables such as (potential) temperature, potential vorticity, or even concentrations of chemical substances (e.g. ozone). One of the most popular definitions is the so-called thermal tropopause according to the WMO (WMO, 1957), which has a some-

what cumbersome wording: *"The first tropopause is defined as the lowest level at which the lapse rate decreases to 2 K per*





*kilometer or less, provided also the average lapse rate between this level and all higher levels within 2 kilometers does not exceed 2 K"* (WMO, 1957). Another definition of a tropopause based on the absolute temperature is the so-called cold-point tropopause (Highwood and Hoskins, 1998), which is widely used for tropical conditions; it is defined as the first minimum of the temperature in the absolute temperature of the free troposphere. Instead of the physical temperature, sometimes the poten-
tial temperature as a quantity conserved under adiabatic processes is used for the determination of the tropopause. For instance, the vertical gradient of potential temperature is used for the determination of the tropopause (Mullendore et al., 2005).

An advantage of these temperature-based tropopause definitions is its ease of use, as a single temperature profile, e.g. from a radio sonde ascent or extracted from model data, is sufficient to determine the tropopause height (Reichler et al., 2003). However, several issues arise from these supposedly simple definitions. Conceptually, they originated from investigations of
synoptic scale flows, assuming clearly pronounced and large vertical gradients in coarse resolution profiles of temperature, and a distinct feature of a single tropopause. In reality, several tropopauses can be found in one profile with this type of definition, especially near the jet stream (Pan et al., 2004).

For dynamic processes in the tropopause region, definitions based on potential vorticity (PV) are often used; this quantity is also conserved under adiabatic processes. However, different PVU values are used in the literature, which are generally
between 2 and 3.5 PVU (Kunz et al., 2011; Hoerling et al., 1991), leading to a kind of indeterminacy in the use of this definition. Finally, it is also possible to define a tropopause on the basis of concentrations of individual chemical substances. In Bethan et al. (1996), for example, a value around $100\,\mathrm{ppb}$ is used to define the ozonopause, which is generally close to the thermal tropopause. Tracer-tracer correlations are also sometimes used for the determination of a tropopause as a transport barrier (Pan et al., 2004). For a more in-depth review of different tropopause definitions the reader is also recommended to the
recent publication of Köhler et al. (2024).

From a more physical perspective, the tropopause structure is determined by the interplay of various different processes on many scales, i.e. adiabatic and diabatic contributions. In the UTLS we find on average synoptic scale motion, i.e. vertical updrafts in the upper troposphere and subsidence in the stratosphere. These vertical motions induce either adiabatic expansion hence cooling of the air (in case of the tropospheric motion), or even adiabatic compression thus warming of air (in case
of stratospheric motion, i.e. the Brewer Dobson circulation, see, e.g., Butchart, 2014). On the other hand, diabatic processes play a role. Water vapor (and also solid water particles, i.e. ice crystals) is absorbing and re-emitting infrared radiation as an almost ideal black body, leading to a local cooling on top of moist layers situated close to the thermal tropopause (Fusina and Spichtinger, 2010). In addition, cloud processes are shaping the thermodynamic structure of the upper troposphere. Ice crystals can form, grow and shrink depending on the relative humidity over ice, which constitutes the thermodynamic control
variable of almost all cloud processes (Pruppacher and Klett, 2010). Finally, friction and irreversible mixing, as e.g. driven by turbulence, contribute to the change in variables as diabatic processes.

Usual definitions of the tropopause, as mentioned above, rely exclusively on adiabatic processes, which can be determined by investigating e.g. the physical or even potential temperature (which is by definition conserved under adiabatic processes). However, from former investigations, it is quite obvious that diabatic processes, especially radiative processes, have a crucial
impact on shaping the tropopause structure; for instance, the so-called tropopause inversion layer (TIL) as a characteristic



feature of the UTLS region is assumed to be crucially created by radiative cooling of water vapor in the tropopause region (Randel et al., 2007). In addition, latent heating from phase changes might introduce cirrus cloud convection thus changed in the vertical UTLS structure (Spichtinger, 2014). In a former study (Köhler et al., 2024), we showed that relative humidity over ice (RHi) is the correct quantity for determining the formation of the TIL, since both aspects (adiabatic and diabatic processes) are included there by definition. Saying that, it is quite natural to adopt a new perspective in terms of using this ideal quantity RHi for a new definition of the tropopause. We propose this new concept of a tropopause definition based on gradients of RHi, which puts the concept of the tropopause as a transport barrier for water vapor to the center of the definition.

Section 2 will introduce the used data as well the description of the new tropopause definition. In Section 3 the performance of the tropopause definition is presented for three cases as well as for a statistical analysis over 10 year. This is followed by theroetical considerations in Section 4.

## 2  Data and Methods

Here we describe the used data and introduce the newly derived tropopause definition based on the gradient of the relative humidity over ice.

### 2.1  Radiosonde data

In this study, we use high-resolution radiosonde data from the German Weather Service (DWD) from the station Meiningen in Central Germany ($50.56173°$N, $10.37693°$E, altitude of $450\,$m a.s.l.), where operational radiosonde ascents are carried out twice a day (00 and 12 UTC). We use the data from 7565 radiosonde ascents between 1st January 2015 to 31st December 2024 (i.e. full 10 years of data). Until 6th September 2017 the measurements were conducted using the Vaisala RS92-SGP sonde. After that the Vaisala RS41-SGP sondes were in operation. Both types are of similar quality and were used before for investigations of the tropopause region (Köhler et al., 2024).

The data has been converted to a regular vertical grid with a resolution of 50 meters. All further examinations were carried out on this grid. The relative humidity over ice is calculated by following equation

$$\text{RHi} = \text{RH} \cdot \frac{p_{s,liq}(T)}{p_{s,ice}(T)} \tag{1}$$

where RH is the relative humidity with respect to liquid (directly measured by the radiosonde), $p_{s,liq}(T)$ is the saturation vapor pressure with respect to liquid water calculated using the Magnus formula, and $p_{s,ice}(T)$ is the saturation vapor pressure with respect to ice obtained by the formula from Murphy and Koop (2005).

### 2.2  Tropopause definition

Two different tropopause definitions are used in this study. The previously widely used WMO definition based on absolute temperature and a newly developed definition based on the gradient of relative humidity with respect to ice. The latter is described first.





### 2.2.1 RHi Gradient Tropopause - RHi-GT

The aim of this definition is to realize a tropopause definition based on the relative humidity over ice. As a final result, it should indicate a troposphere that is significantly more humid than the stratosphere. On the one hand, this could be achieved using a simple threshold value. However, it is very difficult to measure humidity at very cold temperatures, such as those prevailing
in the tropopause region. In order not to be dependent on error-prone absolute values, a gradient method is chosen here. The calculation of the relative humidity over ice gradient tropopause (RHi-GT) is explained in the following:

**Top-down Search:** The algorithm starts at the highest point of the radiosonde ascent and searches downwards for the height, where the following two criteria are met

- **RHi Threshold:** The relative humidity with respect to ice must exceed $10\%$.

- **Vertical Humidity Gradient:** The derivative of RHi with height ($\partial RHi/\partial z$) must be larger than $0.15\%\,\mathrm{m}^{-1}$. This gradient threshold is based on empirical analyses and ensurers that the tropopause height is detected in regions with a strong RHi gradient.

As soon as both criteria are met, the corresponding height is stored as the tropopause height RHi-GT. Note, the RHi-GT algorithm has problems with profiles that show only very small gradients in RHi. It can happen that the criterion for the
gradient only switches on very far down in the profile and delivers unrealistic values. Therefore, a break criterion has been introduced to avoid unrealistically deep tropopause heights lower than $3000\,\mathrm{m}$. In such a case, no value is specified for the tropopause. In a total of 7565 radiosondes, no RHi-GT was found in 42 cases.

### 2.2.2 Thermal tropopause TTP

For the detection of the thermal tropopause the WMO criterion (WMO, 1957) is used. We use therefore the same code as in
Köhler et al. (2024). With this definition of a tropopause, it is also possible that no tropopause height is found. This is the case for 18 radiosonde ascents. The results of both tropopause definitions (red-dotted: thermal tropopause; blue-dashed: RHi-GT) can be seen in Figures 1 to 3.

Note that the high resolution data of temperature raise issues in terms of determining the lapse rate; a smoothing of the profiles is often applied for extracting the large scale feature of thermal profiles (see, e.g., Köhler et al., 2024).

## 3 Results

For our investigations, we define the tropopause via the gradients of relative humidity, as described in section 2, termed relative humidity over ice gradient tropopause (RHi-GT) in the following. Here we present the results of the new RHi-GT definition for the estimation of the tropopause height, based on high-resolution radiosonde data. First, three different real-life situations are shown to demonstrate the capabilities of the RHi-GT definition, also in contrast to the commonly used WMO definition.
The first situation shows a good agreement with the thermal tropopause, while the other two situations show a higher and




lower RHi-GT height, respectively. This points to some caveats of the former definitions, which are overcome by the new one. Furthermore, the distances between the two tropopause definitions during the 10-year observation period are shown. Finally, we present the results relative to the RHi-GT and thermal tropopause, and consider mean profiles.

## 3.1 Comparison of tropopause definitions

To test the performance of the new tropopause definition, we first consider three cases. The first example in Fig. 1 shows a case in which the thermal tropopause and the RHi-GT are close to each other. We see a distinct thermal tropopause with a sharp

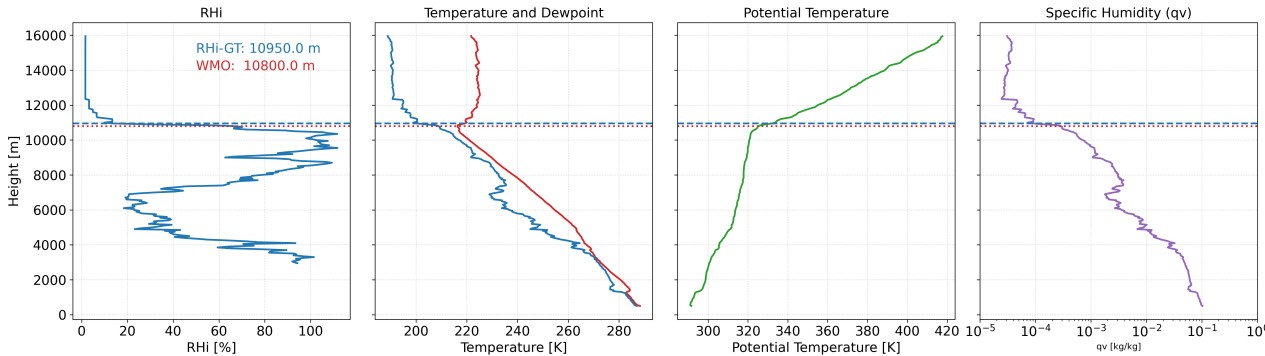

**Figure 1.** Vertical profile of RHi, temperature, dew point, potential temperature and specific humidity of the radiosonde ascent from Meiningen, 2024-06-02, 12 UTC. RHi values are shown for temperatures below $273.15\,\mathrm{K}$. The RHi-GT level (blue-dashed) and thermal tropopause level (red-dotted) are shown in every subplot.

inversion at the tropopause level. The transport barrier is likewise directly visible in the profile of the potential temperature, as a strong positive vertical gradient develops with the tropopause height. Additionally, there is a sharp drop in RHi from very high values above $100\%$ RHi to values below $\mathrm{RHi} \leq 15\%$. The sharp transition (or even transport barrier) is also visible in the

absolute humidity profile.

The second example displays a situation where the RHi-GT is significantly lower than the thermal tropopause. Figure 2 shows a case from 18th February 2024 where the thermal tropopause is calculated at a height of $11.75\,\mathrm{km}$ in contrast to the height corresponding to the RHi-GT definition of $7.0\,\mathrm{km}$. In contrast to the case above (Fig. 1) the signature of the thermal tropopause is not as distinct. In the potential temperature profile a first significant increase of the vertical gradient coincides

with the RHi-GT level at $7.0\,\mathrm{km}$. This means the high RHi values just below the RHi-GT and an increase in stability fall together, while the absolute temperature still decreases for almost $5\,\mathrm{km}$ until the thermal tropopause is reached. At this point the vertical gradient of the $\theta$ is increasing once more. From the RHi-GT point of view, however, we are already deep in the stratosphere, which makes sense with respect to the values of absolute humidity; we see the strong change in absolute humidity occurring at the RHi-GT level, too. In summary, this indicates a strong transport barrier (for water vapor) at the RHi-GT level,

whereas the conventional thermal tropopause definition fails completely to detect this feature.





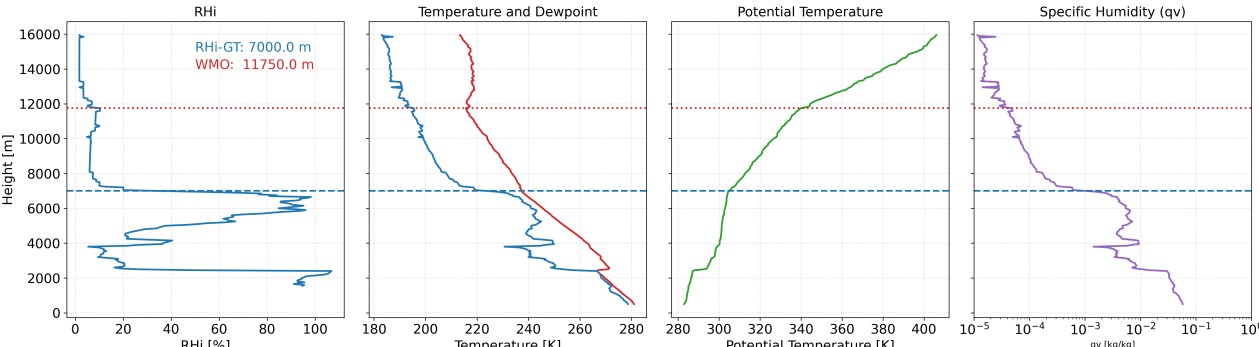

**Figure 2.** Same as Fig. 1, but for 2024-02-18, 00 UTC.

The last example in Fig. 3 exhibits a higher RHi-GT ($9.1\,\mathrm{km}$) compared to the thermal tropopause ($7.9\,\mathrm{km}$). A purely optical determination of the thermal tropopause is difficult here, as the absolute temperature changes only slightly over a deep layer. The WMO thermal tropopause therefore appears somewhat arbitrary at this point. Based on the potential temperature, the tropopause could possibly be placed even lower. However, it is noticeable that a locally strong gradient in the potential

temperature is present at the height of the RHi-GT. This again shows that the RHi-GT definition is also meaningful in terms of stability, and thus indicating a transport barrier for water vapor sufficiently well.

These cases make us confident to use RHi-GT as a new definition for the extratropical tropopause in terms of a transport barrier for water vapor preferred over the conventional thermal tropopause.

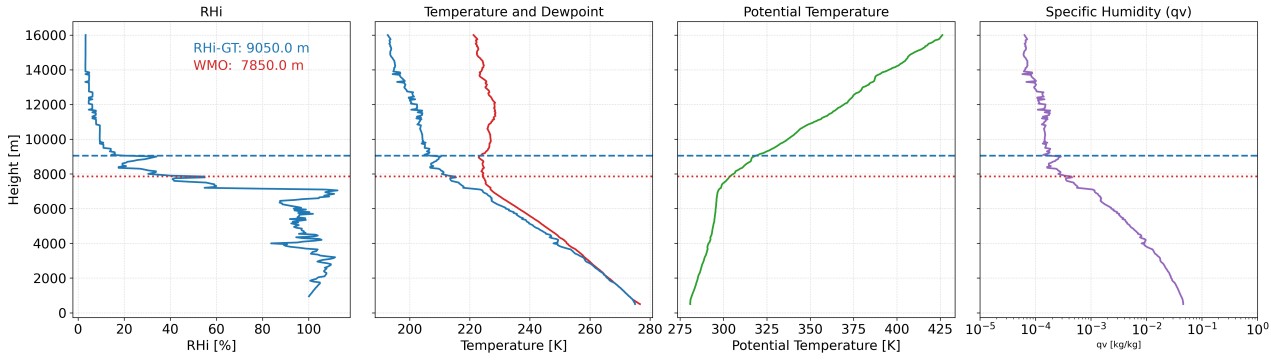

**Figure 3.** Same as Fig. 1, but for 2024-04-20, 00 UTC.

## 3.2 Difference in tropopause height

After considering some individual cases, the differences in the tropopause heights in the period under consideration are now discussed. The histogram in Fig. 4 is designed in a way that negative values indicate a RHi-GT higher than the thermal tropopause. Positive values, on the other hand, indicate a RHi-GT lower than the thermal tropopause, respectively.





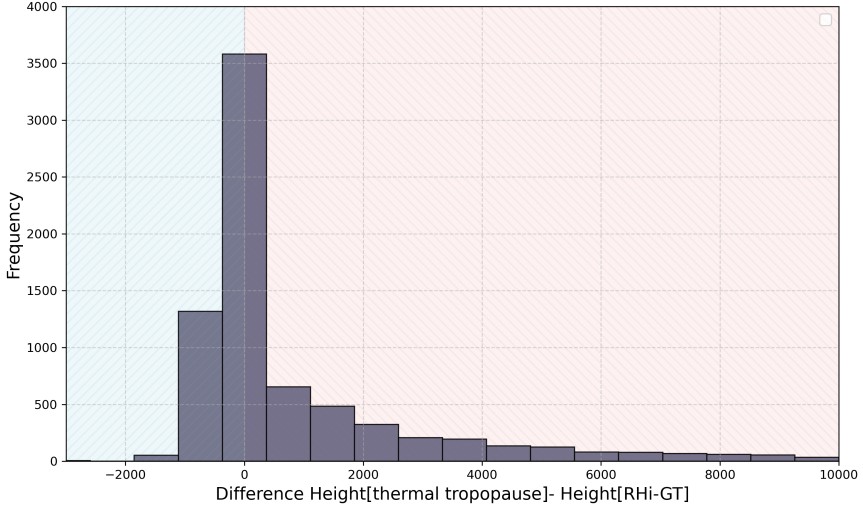

**Figure 4.** Histogram of the height difference in [m] between the thermal tropopause and RHi-GT. Negative values (blue shaded) denote cases, where the RHi-GT altitude is higher than the thermal tropopause. Positive differences (red shaded) denotes cases, where the RHi-GT altitude is below the thermal tropopause.

The results clearly show that most tropopause heights based on RHi-GT are in a similar range than the heights obtained by the WMO criterion. The deviation towards negative height differences, i.e. RHi-GT height is larger than the thermal tropopause, is limited by the dry stratosphere, as the WMO criterion is based on prolonged warming, which does not otherwise occur in the troposphere. Usually, temperature inversions in the lower and mid-troposphere are limited to a narrow layer. On the other hand, the histogram shows for positive difference values an elongated tail, i.e. for cases of RHi-GT heights lower than the thermal tropopause. Especially when the thermal tropopause is quite high, the heights of the two types of tropopause can differ greatly, as for example in Fig. 2, where a high thermal tropopause is found, while the RHi-GT, showing a more physically consistent picture, is about $5\,\mathrm{km}$ lower.

### 3.3 Thermodynamic variables relative to the tropopause

We now investigate some general features of the RHi-GT in comparison to the general definition of the thermal tropopause (TTP). In the following we investigate the variables RHi, absolute temperature, potential temperature and static stability (i.e. the Brunt-Vaisala frequency), respectively. We investigate the full data set of 10 years radiosonde data from Meiningen, Germany. The radiosonde data show great variability between individual ascents and thus reflect the variability of the weather. Therefore, observations of the tropopause are often carried out by means of the transformation to a tropopause-relative height grid (as, e.g., introduced by Birner et al., 2002). This means that the tropopause level is defined as zero, negative altitude values describe the troposphere, and positive values denote the stratosphere; we evaluate the profiles normalized relative to the RHi-GT (blue colors) and the TTP (red colors), respectively.



In Figure 5 the results for RHi are represented. The transformed and averaged profiles of RHi show some distinct features
for the RHi-GT definition (left panel of Figure 5). First, we clearly see the transport barrier for water vapor, appearing as a huge
vertical gradient in RHi by design. There is a strong contrast between the moist troposphere (mean values $\mathrm{RHi} \sim 60\% - 80\%$)
and the very dry stratosphere (values below $\mathrm{RHi} \sim 10\%$), leading to a clear separation of the two spheres, as we would expect
for a measure of the transport barrier. Second, we find enhanced (mean) values of RHi in the troposphere, however close to

the RHi-GT; the mean value has its maximum value at $\mathrm{RHi} \sim 75\%$ just below the tropopause. This is in clear agreement with
former investigations of ice supersaturation in the troposphere; the highest values of RHi (and thus ice supersaturation) are
often found close to the tropopause (Spichtinger et al., 2003; Peter et al., 2006; Reutter et al., 2020). The large spread of the
standard deviation in the troposphere reflects the meteorological variability during the course of 10 years.

The right panel of Fig. 5 shows the result for the widely used thermal tropopause. While the overall picture is similar to the

RHi-GT evaluation, significant differences to the RHi-GT coordinate system are obvious. First, the transition from troposphere
to stratosphere is less pronounced with a broader transition layer. The mean profile shows also a distinctive (and nonphysical)
kink in the RHi curve for the thermal tropopause, while the transition in the RHi-GT case is somewhat smoother. Second, the
mean values of RHi in the troposphere stay at lower values (maximum at $\mathrm{RHi} = 62\%$) and the signature of highest values of
RHi close to the tropopause is not visible.

Finally, it is also noteworthy, that in the RHi-GT case the stratosphere is immediately drier compared to the TTP case.
This is also showing the applicability of RHi-GT as a useful measure for questions concerning water vapor, clouds and, their
radiative impact on climate, since for radiation effects strong vertical gradients lead to the strongest signals (see, e.g., Fusina
and Spichtinger, 2010).

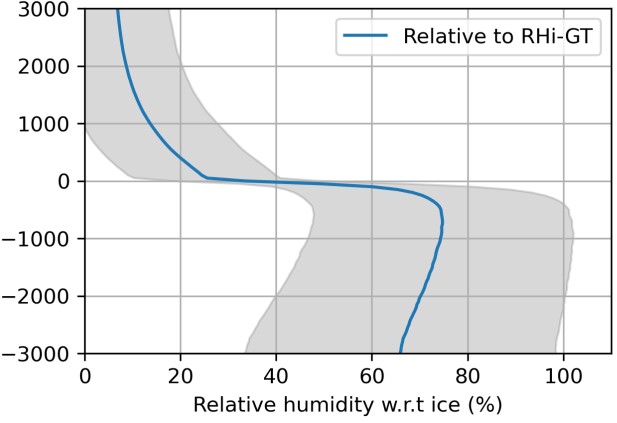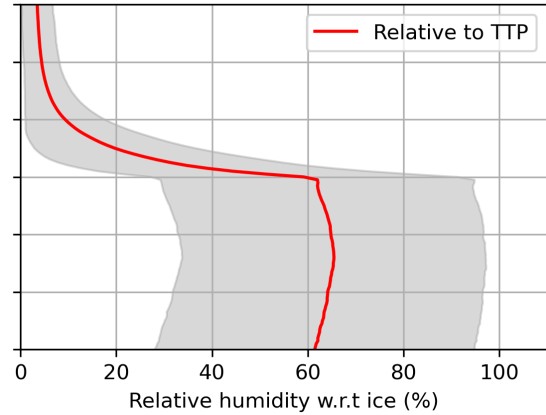

**Figure 5.** Mean of relative humidity over ice for (left) relative to the RHi-GT and (right) relative to the thermal tropopause. Shading denotes
the range of the standard deviation.

In a next step we investigate the (physical) temperature profiles. The absolute temperature profiles are displayed in the left

panel of Figure 6. The profiles for both tropopause types looks quite similar in the troposphere. Just below the height of the RHi-




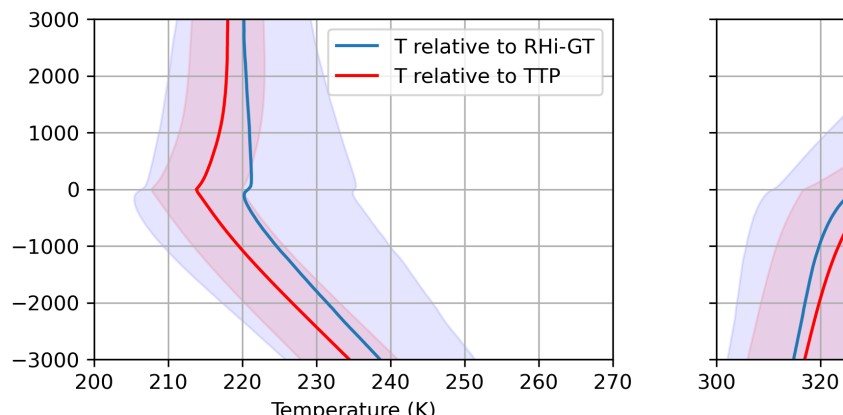
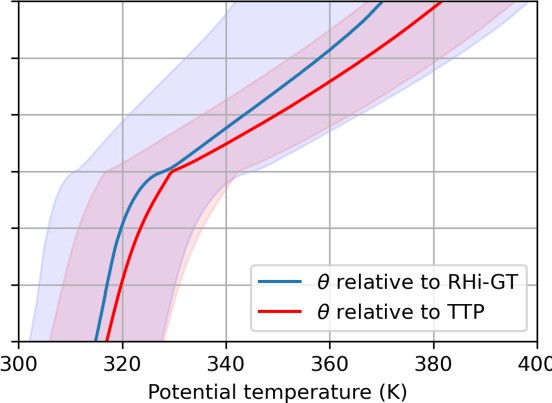

**Figure 6.** Mean of the (left) absolute temperature and (right) potential temperature relative to the tropopause. Blue color denote the results for the RHi-GT, while red colors denote the results relative to the TTP. Shading denotes the range of the standard deviation.

GT a small increase in the absolute temperature is visible, while above the temperature stays more or less constant. In contrast, by design, the results for the thermal tropopause shows the typical minimum of the absolute temperature at tropopause level. Above the thermal tropopause an increase of the temperature can be observed. We clearly see again two features for the RHi-GT in comparison to the classical TTP. First, the mean profile is shifted to higher temperatures; the mean RHi-GT temperature

is about $5\,\mathrm{K}$ higher than the mean TTP temperature. This is due to the fact that in general the RHi-GT is located at lower altitude than the TTP, hence shifting the complete profile. Second, the spread of the temperature is significantly larger for the RHi-GT definition compared to the thermal tropopause definition. This is in accordance with the generally higher variation of RHi-GT tropopause heights, as indicated in figure 4.

The right panel of Fig. 6 shows the potential temperature for both tropopause definitions. Complementary to the shift in

the absolute temperature profile, we see a shift to lower values for the transformed potential temperature profile of RHi-GT compared to TTP; this is consistent with the generally lower levels of RHi-GT tropopause. Similarly, the spread is also higher in the RHi-GT case. However, the potential temperature shows at the tropopause level almost the same values in both averaged profiles. In the qualitative comparison of the profiles in terms of lapse rates we find similar lapse rates in the troposphere; however, for the stratosphere the RHi-GT lapse rate is smaller. Since in case of RHi-GT the transport barrier is related to the

humidity rather than to the thermal structure as for the TTP, this is consistent and meaningful. Finally, the transition between troposphere and stratosphere for the RHi-GT definition is smoother and does not show a bend as it is the case for the TTP definition.

Finally, we investigate the static stability of the tropopause region, measured by the squared Brunt-Väisälä frequency $N^2 := \frac{g}{\theta}\frac{\partial\theta}{\partial z}$. A general feature of stability, and thus a transport obstacle, is the so-called tropopause inversion layer (TIL,

first defined by Birner et al., 2002), which appears for TTP right on top of the tropopause level. The static stability shows a strong increase over a short vertical distance. In contrast to former investigations (Birner et al., 2002; Birner, 2006) we use the




newly defined RHi-GT as the reference point for the coordinate system; both versions (normalized to RHi-GT and TTP) are shown in Figure 7.

The difference of the $N^2$ profile shows clear differences between the tropopause definitions. The $N^2$ profile relative to the

RHi-GT displays a smooth increase from tropospheric values of $N^2 \approx 1 \cdot 10^{-4} \mathrm{s}^{-2}$. However, the vertical gradient of $N^2$ is increasing strongly up to the maximum of $N^2 \approx 8 \cdot 10^{-4} \mathrm{s}^{-2}$ right on the RHi-GT tropopause level. From this point upwards there is a strong and fast decrease to mean values of $N^2 \approx 4 \cdot 10^{-4} \mathrm{s}^{-2}$. Thus, we see a clear separation of the two spheres, with a strong increase in the stability (i.e. the TIL) right at the RHi-GT tropopause level, constituting a strong transport barrier.

The original definition of the TIL using the thermal tropopause shows a more unsteady pattern directly below the thermal

tropopause. While the tropospheric values of the Brunt-Väisälä frequency are similar for both tropopause definitions and thus normalized profiles, a noticeable kink is visible at the local minimum of $N^2 \approx 2 \cdot 10^{-4} \mathrm{s}^{-2}$. This kink can be seen in many studies regarding the TIL (Birner et al., 2002; Gettelman et al., 2011; Köhler et al., 2024). Starting from this kink, $N^2$ rises sharply and then reaches its maximum just above the thermal tropopause with $N^2 \approx 7 \cdot 10^{-4} \mathrm{s}^{-2}$. This is a major difference to the RHi-GT definition, where the height of the $N^2$ maximum is also the tropopause level. Above the $N^2$ maximum the

values of $N^2$ decrease much slower than in the RHi-GT definition. Only $3000\,\mathrm{m}$ above the thermal tropopause similar values of $N^2$ are reached as in the stratospheric part of the RHi-GT definition. The separation is not as pronounced as in the RHi-GT definition.

Thus, we can finally state that a much clearer separation of tropospheric and stratospheric $N^2$ values has been achieved by simply using the gradient of the relative humdity over ice as a measure for the tropopause height.

Overall we see that the property of a transport barrier is much better fulfilled for the new RHi-GT than for other definitions of tropopause levels, which would be a further benefit for investigations of the tropopause region.

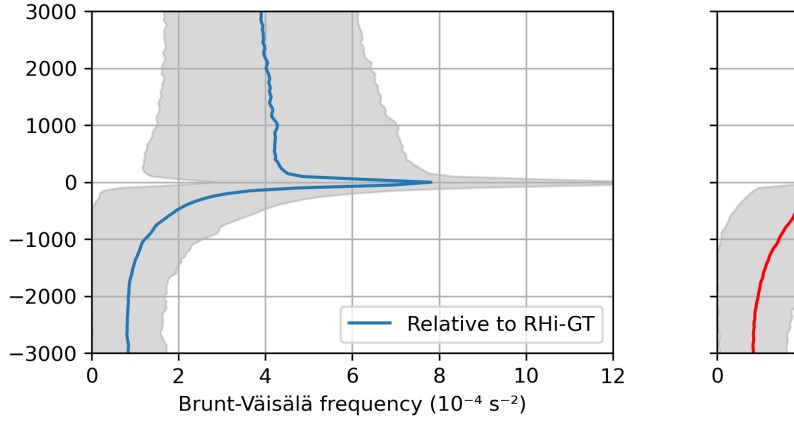
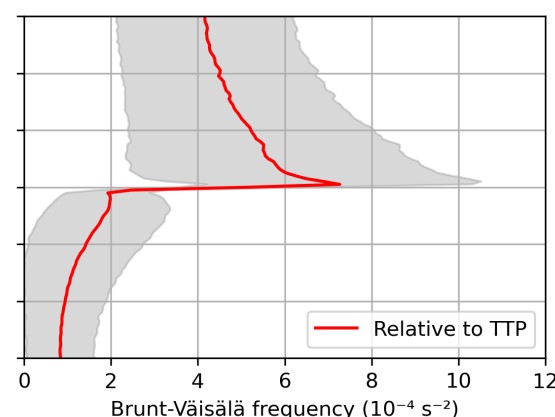

**Figure 7.** Mean of Brunt-Väisälä frequency squared $N^2$ for (left) relative to the RHi-GT and (right) relative to the thermal tropopause. Shading denotes the range of the standard deviation.





## 4 Theoretical considerations

For a better understanding of the processes and a concise diagnostics of the new definition of RHi-GT, we have a look on the total differential of the relative humidity over ice (RHi) and, subsequently, on the (total) derivatives with respect to time and
vertical height, respectively. Using and rewriting the definition of RHi

$$\text{RHi} := 100\% \frac{p_v}{p_{s,ice}(T)} = 100\% \frac{p\, q_v}{\epsilon\, p_{s,ice}(T)} \tag{2}$$

with the partial vapor pressure $p_v$, the saturation vapor pressure over ice $p_{s,ice}(T)$, pressure $p$, temperature $T$, and the absolute humidity $q_v$, respectively. $\epsilon$ denotes the ratio of molar masses of water and air. Assuming water vapor as ideal gas with the specific gas constant $R_v$, and applying the Clausius-Clapeyron equation, we obtain

$$\text{dRHi} = -\text{RHi}\frac{L}{R_v T^2}\text{d}T + \frac{\text{RHi}}{p}\text{d}p + \frac{\text{RHi}}{q_v}\text{d}q_v \tag{3}$$

with the specific latent heat (change in enthalpy) $L$, and the specific gas constant of dry air $R_a$, respectively. This basic equation can be used for (a) investigating the relevant processes for determining the tropopause structure (in terms of time derivatives) and for (b) describing the diagnostics (in terms of spatial derivatives).

First, we consider the time derivatives. We have to distinguish between adiabatic and diabatic processes. In a very good
approximation we can assume that vertical motions (updrafts and downdrafts) induce adiabatic state changes, i.e. expansion or compression (for pressure) of the air parcel producing temperature changes (cooling or warming). For this purpose we can assume that the adiabatic temperature change is given by

$$\frac{\text{d}T}{\text{d}t}\Big|_{\text{adiabatic}} = \frac{\text{d}T}{\text{d}z}\frac{\text{d}z}{\text{d}t} = -\frac{g}{c_p}w, \tag{4}$$

with the vertical motion $w$, and the standard gravity acceleration $g$. In the same spirit using hydrostatic balance (which is
fulfilled to high orders in absence of convection, see, e.g., Klein et al., 2010) and assuming air being an ideal gas with specific gas constant $R_a$, we end up with an equation for the pressure change, i.e.,

$$\frac{\text{d}p}{\text{d}t} = \frac{\text{d}p}{\text{d}z}\frac{\text{d}z}{\text{d}t} = -g\rho w, \quad \frac{\text{RHi}}{p}\frac{\text{d}p}{\text{d}t} = -\text{RHi}\frac{g}{R_a T}w. \tag{5}$$

We have to consider several diabatic processes, i.e. phase changes due to cloud formation and evaporation, radiative processes (absorption and emission of radiation), and friction. The temperature changes can be separated into adiabatic and diabatic
components, i.e.

$$\frac{\text{d}T}{\text{d}t} = \frac{\text{d}T}{\text{d}t}\Big|_{\text{adiabatic}} + \frac{\text{d}T}{\text{d}t}\Big|_{\text{diabatic}}, \tag{6}$$

whereas the diabatic contributions are phase changes, radiation, and friction. The time derivative can be expressed by

$$\frac{\text{dRHi}}{\text{d}t} = \left(\frac{L}{c_p R_v T^2} - \frac{1}{R_a T}\right)\text{RHi}\, gw + \frac{L}{R_v T^2}\text{RHi}\frac{\text{d}T}{\text{d}t}\Big|_{\text{diabatic}} + \frac{p}{\epsilon\, p_{s,ice}(T)}\frac{\text{d}q_v}{\text{d}t}, \tag{7}$$

whereas $\frac{\text{d}q_v}{\text{d}t}$ denotes changes in water vapor, mostly driven by phase changes.

The following conclusions can be drawn from this. The adiabatic term leads to an enhancement of RHi at lower temperatures,



i.e. a homogeneous vertical updraft leads to stronger increase of RHi at higher altitudes/lower temperatures. On the other side, cloud processes lead to a weakening of the humidity gradients: Evaporation and growth of ice crystals change the (relative) humidity towards saturation (thermodynamic equilibrium), i.e. values $\text{RHi} < 100\%$ are enhanced and values $\text{RHi} > 100\%$ are reduced, respectively. Thus, gradients are smeared out within cloud regions, which are usually located below the tropopause.

In essence, cloud processes probably do not affect the determination of the tropopause according to our definition.

Finally, enhanced water vapor concentrations, either inside or outside of clouds, at the tropopause with a strong vertical gradient leads to intense radiative cooling at the top of the layer and thus to sharpening of the thermal gradient (see, e.g., Fusina and Spichtinger, 2010).

If we neglect cloud processes (i.e. phase changes) and the friction term in a first approximation, we end with the following
equation

$$\frac{\mathrm{dRHi}}{\mathrm{d}t} = \left( \frac{L}{c_p R_v T^2} - \frac{1}{R_a T} \right) \text{RHi} gw + \frac{L}{R_v T^2} \text{RHi} \frac{\mathrm{d}T}{\mathrm{d}t} \Big|_{\text{radiation}}, \tag{8}$$

which is later used in a toy model approach.

Second, we use the approach of spatial derivatives (i.e. vertical gradients) for the diagnostics of the tropopause. Using the hydrostatic balance and eq. 3, we derive the different contributions to the vertical gradient of RHi

$$\frac{\partial \mathrm{RHi}}{\partial z} = \underbrace{-\text{RHi} \frac{L}{R_v T^2} \frac{\partial T}{\partial z}}_{\text{temperature}} \underbrace{-\text{RHi} \frac{g}{R_a T}}_{\text{pressure}} + \underbrace{\frac{p}{\epsilon p_{s,ice}(T)} \frac{\partial q_v}{\partial z}}_{\text{humidity}} =: \text{DS}_T + \text{DS}_p + \text{DS}_{q_v} \tag{9}$$

We find that the second term $\text{DS}_p$ is very small, hence does not contribute much to the vertical gradient of RHi. Thus, we find two important contributions to the gradient of RHi, namely contributions from temperature and humidity, respectively.

From this theoretical analysis, we clearly see the huge benefit of our new definition. In all former definitions of the tropopause, only the thermal structure is included, i.e. roughly the first term $\text{DS}_T$ in eq. (9). In the WMO definition, the
temperature gradient over a vertical range is taken into account, maybe also considering changes of temperature stemming from different sources, i.e. adiabatic and diabatic contributiuons. For other definitions, it is even worse. Definitions based on the gradient of potential temperature (Mullendore et al., 2005) or even on absolute values of potential vorticity (Kunz et al., 2011) are neglecting diabatic processes as direct contributions, since both quantities are conserved under adiabatic processes. The use of gradients of relative humidity over ice goes beyond this general procedure, and gives a more precise picture.

For illustrating this added value of considering gradients of RHi, we develop a toy model. We prescribe a relative humidity profile for the upper troposphere and lowermost stratosphere, and the apply adiabatic and diabatic processes in the following way. We use a synoptic vertical updraft ($w = 0.03\,\mathrm{m\,s^{-1}}$) in the UT and a typical radiative cooling contribution at the upper part of the humid layer. For simplification, we completely neglect cloud processes (i.e. the humid layer is cloud free), see discussion above. These forcings are applied for one hour of simulations. The temperature and RHi profiles are showed in Fig. 8, initial
values are represented by blue lines, orange lines show the profiles after a simulation time of one hour.

The gradients of RHi after one hour simulation are showed in Figure 9, the contributions stemming from different terms $\text{DS}_T, \text{DS}_p, \text{DS}_{qv}$ are also displayed.



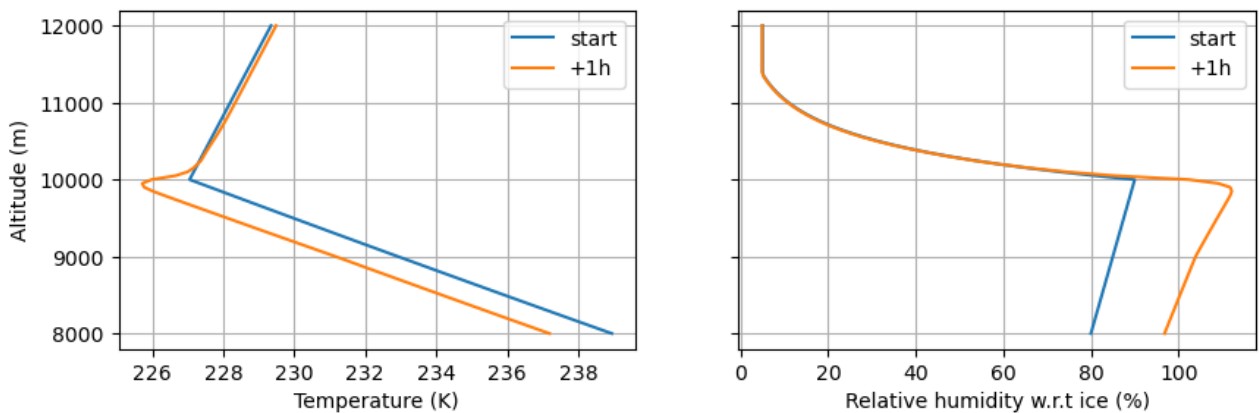

**Figure 8.** Toy model results: Vertical profiles of (left) temperature and (right) relative humidity over ice. Blue lines denote initial conditions and orange lines show the profiles after one hour of simulation.

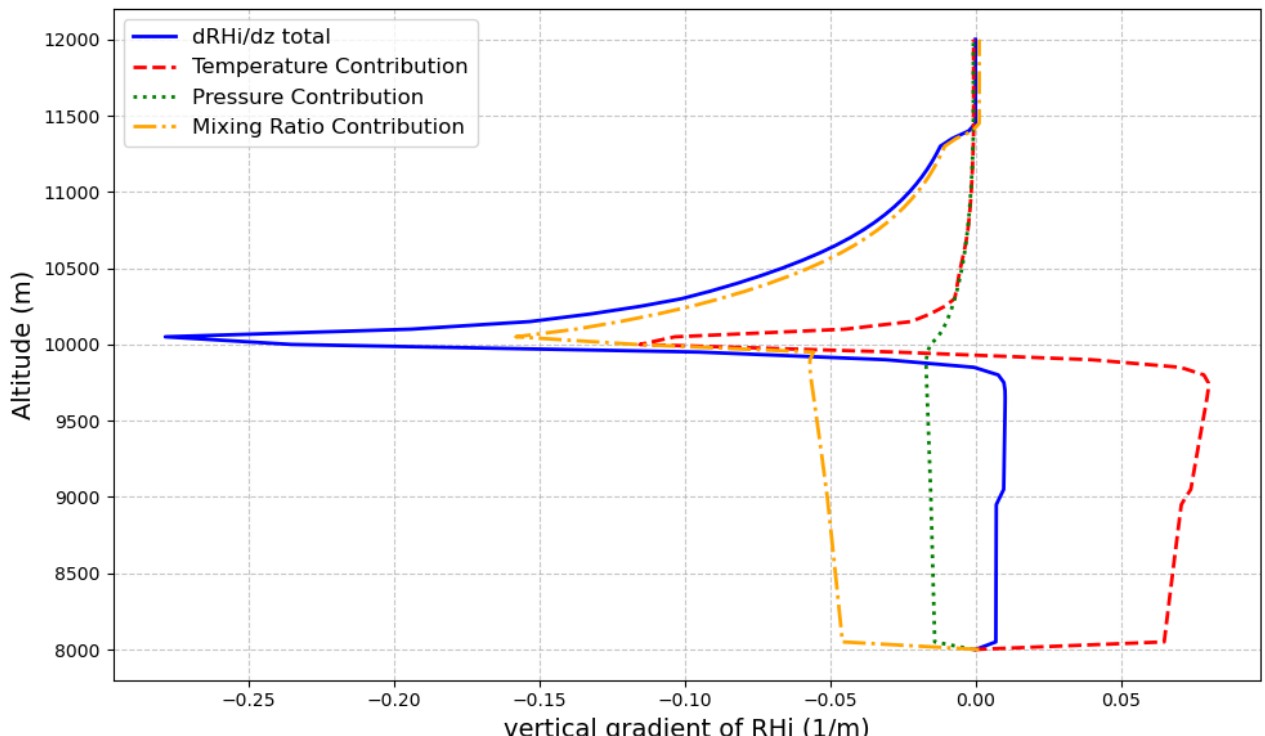

**Figure 9.** Toy model results: Total gradient of RHi as well as contributions from individual quantities.

From this Figure, it is very obvious that the major contributions for the strong gradient in RHi stem from temperature and humidity terms. The superposition of the different contributions lead to a very distinct maximum in the gradient, which can be



used as a definition for the tropopause, and this is exactly our approach of the new definition RHi-GT. Finally, the values of the mean peak are close to the used threshold for the search algorithm in sec. 2.

## 5  Summary

We present a novel approach for a tropopause definition based on the gradient of relative humidity with respect to ice (RHi). Based on high-resolution radio sonde data from Meiningen, Germany (2015-2024), we can show that the so-called RHi-GT

(RHi Gradient Tropopause) definition better reflects the nature of the tropopause as a transport barrier compared to the widely used WMO thermal tropopause. Determining the tropopause height with the help of RHi-GT is very simple. Starting at the highest available level of a radiosonde, the first gradient in RHi that exceeds a value of $0.15\,\%\,\mathrm{m}^{-1}$ is sought downwards, while the RHi itself must be at least above $10\%$. In individual profiles from three different situations, the RHi-GT tropopause aligns better with signs of a transport barrier, such as sharp gradients in absolute humidity or increases in static stability. When

analyzing thermodynamic variables over 10 years relative to the RHi-GT tropopause height it evident that RHi and static stability ($N^2$) show a much sharper and clearer transition between troposphere and stratosphere directly at the tropopause level compared to the thermal tropopause. A theoretical analysis shows that the vertical gradient of RHi primarily results from temperature and humidity contributions, which is supported by a simple toy model.

In conclusion, we can state that the new definition of a tropopause determined from gradients of relative humidity over ice,

i.e. the RHi-GT, puts the focus on the transport barrier for water vapor; in this respect it outperforms the usual definitions relying on the thermal structure of profiles only, and is very well suited for qualitative and quantitative investigations of the tropopause region, especially including the relevant diabatic processes, as e.g. radiation and phase changes, respectively.

*Code availability.*  Code for the data processing and analysis is provided here: https://figshare.com/s/ca1d0c5b8a38e3fb7fa8.

*Data availability.*  The radiosonde data are publicly available from https://opendata.dwd.de/ at DWD.

*Author contributions.*  PR and PS designed the study; PR carried out the data analyses; PS developed and run the toy model; PR and PS contributed to interpreting the results and writing the paper.

*Competing interests.*  The contact author has declared that none of the authors has any competing interests.





*Acknowledgements.* We thank Deutscher Wetterdienst (DWD) for providing the high resolution radiosonde data. Philipp Reutter acknowledges support by the DFG within the Transregional Collaborative Research Centre TRR301 TPChange, Project-ID 428312742 , project C01.
Peter Spichtinger acknowledges support by the DFG within the Transregional Collaborative Research Centre TRR301 TPChange, Project-ID 428312742, project B07.



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
