# Peer review of "The Frosty Frontier: Redefining the Tropopause as a transport barrier using the Relative Humidity over Ice"

_EGUsphere, 2025_

## Referee Comment (RC1)

**Review of "The Frosty Frontier: Redefining the Tropopause as a transport barrier using the Relative Humidity over Ice "**

BY PHILIPP REUTTER AND PETER SPICHTINGER

**General**

This paper reports on a novel way of deducing a tropopause using the relative humidity over ice (RHi). This is good. The tropopause defined in this way is analyzed as a transport barrier between the stratosphere and the troposphere. It is pointed out that RHi is the key variable for ice cloud formation, which incorporates both diabatic and adiabatic processes. The analysis focuses on a ten year record of measurements at a radio sonde station (Meiningen); I like the statistical comparison of the data in Fig. 4.

My reservations regarding the paper are: first, the paper discusses on the tropopause in Northern hemisphere mid-latitudes. Clearly, the tropopause in the tropics is different and the concepts presented here may not carry over (see also below). Second, this analysis focuses in the role of the tropopause as a transport barrier for water vapor. Water vapor is an important compound, but not the only one. My suggestion is that these two points should be made very clear in the abstract and in the title of a revised version of the paper.

Further , the comparison in the manuscript is only with the classic WMO lapse rate tropopause. I agree it is widely used, but it is not the only "tropopause" (see also below). In particular, the lapse rate tropopause and the cold point tropopause are not the same quantity. Also here the paper could be more explicit.

Finally, the manuscript stresses the importance of diabatic processes for determining the tropopause – this is good. But I think the manuscript is unnecessarily vague by simply saying "diabatic". The diabatic processes in question are radiation (but on which time scales) and cloud formation (but phase changes are neglected in the analysis in Sec. 4). These points should be clearer in the manuscript.

Overall, I think there is a lot of good and interesting material here, but I am afraid the manuscript needs work before it can be accepted for ACP.

**Detail**

**Definition of the tropopause**

I know that the wording is widely used in the literature, but I do not like the expression of a "definition of the tropopause". As the authors say, "the tropopause acts as transport barrier between the troposphere and stratosphere." – so the ambition has to be finding a way of *locating the barrier* not simply *defining* it. It looks to me that this issue is not taken into account here. I wanted to bring up the issue of a "definition of the tropopause" but this is a choice of the authors.

**The tropical and the extra-tropical tropopause**

Aren't the tropical and the extra-tropical tropopause two rather different entities? Many papers have discussed this question (e.g, Highwood and Hoskins, 1998; Fueglistaler et al., 2009; Bian et al., 2012; Pan et al., 2018; Krämer et al., 2020; Hoffmann and Spang, 2022). I think this aspect should not be ignored here – to me the discussion has rather a focus on the extra-tropical conditions. Definitely, the high resolution radio sonde data are only available for mid-latitude conditions. If the authors agree, it would be good to state this explicitly throughout the manuscript and even include the information in the title.

**The lapse rate and the cold point tropopause**

The lapse rate and the cold point tropopause are treated here as two alternative "definitions" of the tropopause. But I would argue that they are not describing the same transport barrier. Pan et al. (2018) and Munchak and Pan (2014) made the point that the lapse rate tropopause and the cold point tropopause are not the same thing. (These studies are for the tropics so perhaps they are not applicable here – but then the question remains how the two tropopauses differ in mid-latitudes.) For example, transport of water vapor should be differently affected by the lapse rate and the cold point tropopause. Further information could be found in König et al. (2019). Convective transport across the lapse rate tropopause is much more common than water vapor transport across the cold point tropopause (Clapp et al., 2023).

In general, for different trace gasses, not necessarily the same tropopause is relevant (Bauchinger et al., 2025). Finally, to play devil's advocate: how different

would the tropopause look if simply water vapor was considered instead of relative humidity over ice (see also below)?

Moreover, in the manuscript, the term "thermal tropopause" and "lapse rate tropopause" (where the classic WMO definition uses the term lapse rate) mean the same thing. This could be explicitly explained in the manuscript.

**Literature**

Overall, the impression of the reviewer is that the coverage of the existing literature on the tropopause is better in the previous paper by some of the authors (Köhler et al., 2024, K24 hereafter), than in the present manuscript. For example the classic study by Hoinka (1997) is discussed in K24 but not here. The same is true for the study by Maddox and Mullendore (2018), who point out that there are potentially two different interpretations of the classic WMO lapse rate definition. Further, using potential vorticity (PV) as an entity to be considered when deducing a tropopause is criticized by the argument that choosing a particular PV value introduces a certain arbitrariness; however, when the maximum *gradient* of PV is considered, this is not the case (Kunz et al., 2011; Turhal et al., 2024).

**Transport barrier for certain compounds**

The characteristics of the tropopause as a transport barrier might be different for different compounds (e.g., Pan et al., 2018; Bauchinger et al., 2025). From the manuscript it seems that here the focus is on a transport barrier for water vapor (e.g. l. 67). If this is correct, this fact should be obvious in the abstract (and perhaps the title). If this is not correct the manuscript should demonstrate in how far the new tropopause definition is relevant for other trace species (say $N_2O$ or $CH_4$).

**Minor points**

- l. 6: "approaches": only a comparison with the thermal tropopause is presented here

- l. 13: Looks like the entire study is on the extra-tropics

- l. 17: Note that STE (in particularly of $H_2O$) is often only poorly represented in models (Charlesworth et al., 2023).

- Further studies of the vertical gradient of potential temperature as a measure of the tropopause (Kohma and Sato, 2019).

- l. 36: I do not agree. Already the old WMO lapse rate definition considers two tropopauses. It should be clear here if the classic double tropopause issue is meant here (e.g., Randel et al., 2007; Castanheira and Gimeno, 2011) or something else.

- l. 40: The PV values attributed to the location of the tropopause are to some extent arbitrary, I agree. However this is not the case for more recent studies (e.g. Kunz et al., 2011; Turhal et al., 2024), where the maximum gradient of PV is considered.

- l. 47 ff.: This paper states "In the UTLS we find on average synoptic scale motion, i.e. vertical updrafts in the upper troposphere and subsidence in the stratosphere. These vertical motions induce either adiabatic expansion hence cooling of the air (in case of the tropospheric motion), or even adiabatic compression thus warming of air (in case of stratospheric motion, i.e. the Brewer Dobson circulation..." – I am confused here. Clearly the Brewer Dobson circulation is associated with ascending motion in the tropics. This seems in contrast to what is stated here. – Please clarify.

- l. 49: ice particles are also important for short wave radiation.

- l. 50: it is not clear here why the moist layer should be close to the thermal tropopause (where thermal tropopause is the lapse tropopause if I read the paper correctly)

- l. 59: citations for "former investigations"?

- l. 59: Aren't the time scale relevant here? Radiative processes are surely important, but what are the timescales in question here? The location of the tropopause is valid only for a given time span, isn't it?

- l. 60: one could mention Birner's papers (e.g., Birner et al., 2002, 2006) for the TIL;

- l. 64: There are a lot of diabatic processes of relevance to the atmosphere so it is not sufficient to include diabatic processes in general in the analysis. Aren't the *right* processes important here? And this is again a time scale issue.

- l. 71: vertical gradient

- l. 81: What is the advantage of choosing a regular grid?

- l. 84: "directly measured" – how good is the measurement at low temperature and low water vapor? Can this be quantified? Citations?

- l. 85: citation for the Magnus formula? Is there an assessment of the quality of this formula?

- l. 86. Is there an assessment of the quality of the Murphy and Koop formula?

- Sec. 2.2: Would not the hygropause also be a valid/important comparison?

- l. 94: threshold for water vapor

- l. 99: how is this threshold chosen?

- l. 104: a small gradient in RHi means a small gradient in water vapor – correct? How can this happen if the troposphere is humid an the stratosphere is dry? Is this an indication that the transport barrier is not very pronounced under these circumstances?

- l. 107: report 42 also in percent.

- l. 111: Only 18 "no tropopause cases" vs. 42 (l. 107); report 18 also in percent. Is 18 "better" than 42?

- Figures 1-3: How are these cases selected?

- Fig. 1, caption: state what is temperature (red line I guess) and what is dew point in panel b. The sonde is labeled 12 UTC but the measurements were taken over a certain time period. Report the period.

- l. 121: why "definitions" – you only compare with the thermal tropopause

- l. 121: "caveats": this is not completely obvious, the figures indicate that the results are different, not which tropopause is better. Also note that the cold point and the lapse rate tropopause are often not co-located as well.

- l. 122: shown – where?

- l. 130: which panel shows the "absolute humidity profile"?

- l. 137: "the $\theta$" does not sound right

- l. 142; "optical"?

- l. 146: The implicit assumption seems here that the upward transport of water vapor (and the freeze out) occurs along the vertical profile. Of course, this is an over-simplification. To understand the water vapor at 10 km in (say) Fig. 2 one needs to consider the Lagrangian history of the air parcel (e.g., Schiller et al., 2009; Khaykin et al., 2022), not only what happens below.

- ls. 147-148: this "confidence" depends very much on the question how the cases in Figs. 1-3 are selected.

- Fig. 4: I like this statistical comparison in this Figure. Nonetheless, the reference tropopause is only the thermal tropopause here as throughout the paper.

- l. 155: "prolonged warming" is unclear; explain and/or provide references.

- l. 169: which Figures are these colors referring to?

- l. 170: results employing RHi for the tropopause determination

- l. 177: Would Krämer et al. (2020) also be a suitable citation here?

- l. 177: What means "close" here? Approximate value in km?

- l. 171: why is the kink nonphysical? In individual ascents (Figs. 1-3) there seem to be also kinks

- l. 187: "signals" in which quantity? Also, does the radiation care about the tropopause? And means relative to a tropopause height?

- l. 195/196: "lower altitude" – should this be visible in Fig. 4? "lower altitude" seems not obvious from Fig. 4. Suggest some discussion.

- l. 223: can you quantify "just above" (in m)?

- l. 230: "definitions": here only a comparison o the thermal tropopause is presented

- l. 241: I cannot find $R_a$ in eq. 3

- l, 249: mention $c_p$

- l. 257, eq. (7): The time derivative of RHi (using eq. 3) can be expressed ...(suggest being a bit more explicit here)

- l. 261: note that the updraft is not necessarily vertical.

- l. 264: Yes, clouds are usually located below the tropopause, but not always. For example short term cooling could be induced by upward propagating gravity waves. Could this aspect be of relevance here?

- l. 265: Is this true also for the situation within a convective cloud, where a tropopause is hard to find (Maddox and Mullendore, 2018).

- l. 267: which "layer"?

- l. 269: I think it would be easier to follow these considerations if it is better explained why the specific latent heat $L$ is relevant here if phase changes are neglected.

- l. 275: Eq 9: I am not sure here: Is this equation true in general or is it formulated neglecting cloud processes?

- l. 276: why is the second term small?

- l. 278: "all" is not applicable here – the comparison in the arguments presented here is comparsion to the thermal tropopause. For example, if an ozone tropopause is considered (Hoinka, 1997), the thermal structure of the tropopause is not relevant.

- l. 283: "diabatic processes" are put forward here; I think the paper could make a stronger point if it states which processes are relevant here. First, I think that friction is neglected. Second, I think that radiative time scales are too slow (do the authors agree?). That leaves phase change (cloud formation) as the central process. If the authors agree, these arguments could be more explicit.

- l. 286: change "the apply"

- l. 287: What are the typical radiative time scales in question here?

- l. 287: How is $w$ determined? Citation?

- For the toy model, also the assumption of a temperature profile is necessary, if I understand correctly – how is it determined?

- l. 289 and 291: "showed" is not correct

- Fig. 8, left: the temperature decrease seems to be throughout the troposphere (as far as visible in the figure), more than 1 K per hour. How realistic are such cooling rates?

- l. 306: This analysis assumed the absence of cloud formation .. this should be explicitly stated.

- l. 312: but phase changes are neglected in Section 4 as far as I can see. I think the paper could be more explicit about the relative importance of radiation and phase changes regarding how a tropopause location can be determined.

- l. 330: journal title should not be in capitals

- l. 334: why is "atmosphere" in capitals?

- l. 342: there should be no space in the formula of $H_2O$

**References**

Bauchinger, S., Engel, A., Jesswein, M., Keber, T., Bönisch, H., Obersteiner, F., Zahn, A., Emig, N., Hoor, P., Lachnitt, H.-C., Weyland, F., Ort, L., and Schuck,

T. J.: The extratropical tropopause – Trace gas perspective on tropopause definition choice, EGUsphere, 2025, 1–29, doi:10.5194/egusphere-2025-1589, url: https://egusphere.copernicus.org/preprints/2025/egusphere-2025-1589/, 2025.

Bian, J., Pan, L. L., Paulik, L., Vömel, H., and Chen, H.: In situ water vapor and ozone measurements in Lhasa and Kunmin during the Asian summer monsoon, Geophys. Res. Lett., 39(19), L19808, doi:10.1029/2012GL052996, 2012.

Birner, T., Dörnbrack, A., and Schumann, U.: How sharp is the tropopause at midlatitudes?, Geophys. Res. Lett., 29(14), 2002.

Birner, T., Sankey, D., and Shepherd, T. G.: The tropopause inversion layer in models and analyses, Geophys. Res. Lett., 33(L14804), 2006.

Castanheira, J. M. and Gimeno, L.: Association of double tropopause events with baroclinic waves, J. Geophys. Res., 116(D19113), doi:10.1029/2011JD016163, 2011.

Charlesworth, E., Ploeger, F., Birner, T., Baikhadzhaev, R., Abalos, M., Abraham, L., Akiyoshi, H., Bekki, S., Dennison, F., Jöckel, P., Keeble, J., Kinnison, D., Morgenstern, O., Plummer, D., Rozanov, E., Strode, S., Zeng, G., and Riese, M.: Stratospheric water vapor affecting atmospheric circulation, Nat. Commun., 14, 3925, url: https://doi.org/10.1038/s41467-023-39559-2, 2023.

Clapp, C. E., Smith, J. B., Bedka, K. M., and Anderson, J. G.: Distribution of cross-tropopause convection within the Asian monsoon region from May through October 2017, Atmos. Chem. Phys., 23(5), 3279–3298, doi:10.5194/acp-23-3279-2023, url: https://acp.copernicus.org/articles/23/3279/2023/, 2023.

Fueglistaler, S., Dessler, A. E., Dunkerton, T. J., Folkins, I., Fu, Q., and Mote, P. W.: Tropical tropopause layer, Rev. Geophys., 47, RG1004, doi: 10.1029/2008RG000267, 2009.

Highwood, E. J. and Hoskins, B. J.: The tropical tropopause, Q. J. R. Meteorol. Soc., 124, 1579 – 1604, 1998.

Hoffmann, L. and Spang, R.: An assessment of tropopause characteristics of the ERA5 and ERA-Interim meteorological reanalyses, Atmos. Chem. Phys., 22(6), 4019–4046, doi:10.5194/acp-22-4019-2022, url: https://acp.copernicus.org/articles/22/4019/2022/, 2022.

Hoinka, K. P.: The tropopause: discovery, definition and demarcation, Meteorol. Z., 6(6), 281–303, 1997.

Khaykin, S. M., Moyer, E., Krämer, M., Clouser, B., Bucci, S., Legras, B., Lykov, A., Afchine, A., Cairo, F., Formanyuk, I., Mitev, V., Matthey, R., Rolf, C., Singer, C. E., Spelten, N., Volkov, V., Yushkov, V., and Stroh, F.: Persistence of moist plumes from overshooting convection in the Asian monsoon anticyclone, Atmos. Chem. Phys., 22(5), 3169–3189, doi:10.5194/acp-22-3169-2022, url: https://acp.copernicus.org/articles/22/3169/2022/, 2022.

Köhler, D., Reutter, P., and Spichtinger, P.: Relative humidity over ice as a key variable for Northern Hemisphere midlatitude tropopause inversion layers, Atmos. Chem. Phys., 24(17), 10 055–10 072, doi:10.5194/acp-24-10055-2024, url: https://acp.copernicus.org/articles/24/10055/2024/, 2024.

Kohma, M. and Sato, K.: A Diagnostic Equation for Tendency of Lapse-Rate-Tropopause Heights and Its Application, J. Atmos. Sci., 76(11), 3337 – 3350, doi:10.1175/JAS-D-19-0054.1, url: https://journals.ametsoc.org/view/journals/atsc/76/11/jas-d-19-0054.1.xml, 2019.

König, N., Braesicke, P., and von Clarmann, T.: Tropopause altitude determination from temperature profile measurements of reduced vertical resolution, Atmos. Meas. Tech., 12(7), 4113–4129, doi:10.5194/amt-12-4113-2019, url: https://amt.copernicus.org/articles/12/4113/2019/, 2019.

Krämer, M., Rolf, C., Spelten, N., Afchine, A., Fahey, D., Jensen, E., Khaykin, S., Kuhn, T., Lawson, P., Lykov, A., Pan, L. L., Riese, M., Rollins, A., Stroh, F., Thornberry, T., Wolf, V., Woods, S., Spichtinger, P., Quaas, J., and Sourdeval, O.: A microphysics guide to cirrus – Part 2: Climatologies of clouds and humidity from observations, Atmos. Chem. Phys., 20(21), 12 569–12 608, doi:10.5194/acp-20-12569-2020, url: https://acp.copernicus.org/articles/20/12569/2020/, 2020.

Kunz, A., Konopka, P., Müller, R., and Pan, L. L.: Dynamical tropopause based on isentropic potential vorticity gradients, J. Geophys. Res., 116, D01110, doi:10.1029/2010JD014343, 2011.

Maddox, E. M. and Mullendore, G. L.: Determination of Best Tropopause Definition for Convective Transport Studies, J. Atmos. Sci., pp. 3433–3446, url: https://doi.org/10.1175/JAS-D-18-0032.1, 2018.

Munchak, L. A. and Pan, L. L.: Separation of the lapse rate and the cold point tropopauses in the tropics and the resulting impact on cloud top-tropopause relationships, J. Geophys. Res., 119(13), 7963–7978, doi:https://doi.org/10.1002/2013JD021189, url: https://agupubs.onlinelibrary.wiley.com/doi/abs/10.1002/2013JD021189, 2014.

Pan, L. L., Honomichl, S. B., Bui, T. V., Thornberry, T., Rollins, A., Hintsa, E., and Jensen, E. J.: Lapse Rate or Cold Point: The Tropical Tropopause Identified by In Situ Trace Gas Measurements, Geophys. Res. Lett., 45(19), 10 756–10 763, doi:10.1029/2018GL079573, url: https://agupubs.onlinelibrary.wiley.com/doi/abs/10.1029/2018GL079573, 2018.

Randel, W. J., Seidel, D. J., and Pan, L. L.: Observational characteristics of double tropopauses, J. Geophys. Res., 112, D07309, doi:10.1029/2006JD007904, 2007.

Schiller, C., Grooß, J.-U., Konopka, P., Plöger, F., Silva dos Santos, F. H., and Spelten, N.: Hydration and dehydration at the tropical tropopause, Atmos. Chem. Phys., 9(24), 9647–9660, doi:10.5194/acp-9-9647-2009, 2009.

Turhal, K., Plöger, F., Clemens, J., Birner, T., Weyland, F., Konopka, P., and Hoor, P.: Variability and trends in the potential vorticity (PV)-gradient dynamical tropopause, Atmos. Chem. Phys., 24(23), 13 653–13 679, doi:10.5194/acp-24-13653-2024, url: https://acp.copernicus.org/articles/24/13653/2024/, 2024.

---

## Author Comment (AC1)

**Reply to Reviewer**

General comment:

We would like to thank both reviewers for their many helpful comments and suggestions, which significantly improved the manuscript. The reviewers' time and effort are greatly appreciated. We changed some of the wording, especially regarding the tropopause as a transport barrier. We removed the section with the toy models, since it was not clearly formulated and, in addition, an exhaustive investigation of tropopause related processes using the model is beyond of the scope of this investigation. Now, the focus of the study lies on the observational investigation of a tropopause based on gradients of relative humidity with the addition of an investigation into the sensitivity of the parameters for defining the RHi-GT tropopause

In the following, we answer to the comments point by point. Questions and remarks of the reviewers are marked in orange, reply of the authors are marked in black and changes to the manuscript are marked in blue.

Reviewer #1

My reservations regarding the paper are: first, the paper discusses on the tropopause in Northern hemisphere mid-latitudes. Clearly, the tropopause in the tropics is different and the concepts presented here may not carry over (see also below). Second, this analysis focuses in the role of the tropopause as a transport barrier for water vapor. Water vapor is an important compound, but not the only one. My suggestion is that these two points should be made very clear in the abstract and in the title of a revised version of the paper.

We agree that the term 'transport barrier' is too strong a statement in the title. We have therefore changed the title to focus on the mid-latitudes, for which our study is exemplary. The new title is now:

*The Frosty Frontier: Redefining the mid-latitude Tropopause using the Relative Humidity over Ice*

Further, the comparison in the manuscript is only with the classic WMO lapse rate tropopause. I agree it is widely used, but it is not the only "tropopause" (see also below). In particular, the lapse rate tropopause and the cold point tropopause are not the same quantity. Also here the paper could be more explicit.

We agree. We have clarified in the manuscript that our study focuses on the mid-latitudes. In this context, the thermal (lapse rate) tropopause serves as a commonly used reference, which allows us to compare our RHi-GT-based tropopause definition.

The primary aim of this paper is to document the observation of an RHi-GT tropopause. We recognize that other definitions, such as the cold-point tropopause, are also important. A more detailed comparison with different tropopause definitions across other regions, as well as a

closer look at the processes that lead to the formation of RHi-GT, will be the focus of future studies.

Finally, the manuscript stresses the importance of diabatic processes for determining the tropopause – this is good. But I think the manuscript is unnecessarily vague by simply saying "diabatic". The diabatic processes in question are radiation (but on which time scales) and cloud formation (but phase changes are neglected in the analysis in Sec. 4). These points should be clearer in the manuscript.
Overall, I think there is a lot of good and interesting material here, but I am afraid the manuscript needs work before it can be accepted for ACP.
Thank you very much for your input, we will answer this within in the detailed issues below.

Detail

Definition of the tropopause

I know that the wording is widely used in the literature, but I do not like the expression of a "definition of the tropopause". As the authors say, "the tropopause acts as transport barrier between the troposphere and stratosphere." – so the ambition has to be finding a way of locating the barrier not simply defining it. It looks to me that this issue is not taken into account here. I wanted to bring up the issue of a "definition of the tropopause" but this is a choice of the authors.
We can understand the reluctance to use the term 'definition'. However, we believe that our approach to identifying the tropopause using a strong vertical RHi gradient fulfils the requirements for using the word „definition", as we clearly describe how we believe this version of the tropopause should be identified.
In future studies, we will address the transport barrier for water vapor and other trace gases.

The tropical and the extra-tropical tropopause Aren't the tropical and the extra-tropical tropopause two rather different entities? Many papers have discussed this question (e.g, Highwood and Hoskins, 1998; Fueglistaler et al., 2009; Bian et al., 2012; Pan et al., 2018; Kr¨ amer et al., 2020; Hoffmann and Spang, 2022). I think this aspect should not be ignored here – to me the discussion has rather a focus on the extra-tropical conditions. Definitely, the high resolution radio sonde data are only available for mid-latitude conditions. If the authors agree, it would be good to state this explicitly throughout the
manuscript and even include the information in the title.
You are absolutely right and we have revised the manuscript to make it clear that we are only looking at the tropopause in the mid-latitudes. We made this clear first of all in the title, which we changed to
*The Frosty Frontier: Redefining the mid-latitude Tropopause using the Relative Humidity over Ice*
For the time being, we will restrict ourselves to mid-latitudes, as the tropical tropopause and the arctic tropopause also differ structurally from the tropopause in the mid-latitudes (see e.g. Zängl & Hoinka, 2001, Hoffmann & Spang, 2022).

Our aim with this manuscript is to present the idea of a tropopause definition based on the RHi gradient. We think that this works very well for the mid-latitudes, but for the application in the polar regions as well as in the tropics a thorough investigation is certainly necessary, which is beyond the scope of this manuscript.

We have made the restriction to the mid-latitudes clear in the manuscript, especially in the introduction.

The lapse rate and the cold point tropopause are treated here as two alternative "definitions" of the tropopause. But I would argue that they are not describing the same transport barrier. Pan et al. (2018) and Munchak and Pan (2014) made the point that the lapse rate tropopause and the cold point tropopause are not the same thing. (These studies are for the tropics so perhaps they are not applicable here – but then the question remains how the two tropopauses differ in mid-latitudes.) For example, transport of water vapor should be differently affected by the lapse rate and the cold point tropopause. Further information could be found in König et al. (2019). Convective transport across the lapse rate tropopause is much more common than water vapor transport across the cold point tropopause (Clapp et al., 2023). In general, for different trace gasses, not necessarily the same tropopause is relevant (Bauchinger et al., 2025). Finally, to play devil's advocate: how different would the tropopause look if simply water vapor was considered instead of relative humidity over ice (see also below)?

We fully agree that different tropopause definitions, such as the lapse rate and cold-point tropopause, do not necessarily describe the same transport barrier, and their impact on trace gas transport may differ. In our study, we focus on the mid-latitudes and use the thermal (lapse rate) tropopause as a well-established reference to compare with our RHi-GT-based tropopause. The main aim here is to document the occurrence of an RHi-GT tropopause; we do not yet explore the implications for transport processes. We agree that comparing different tropopause definitions for various trace gases and across different regions would be an important task for future studies. Likewise, looking into how a water-vapor-based tropopause compares to the RHi-GT tropopause is an interesting direction for follow-up work.

The relative humidity depends linearly on water vapor mixing ratio. For almost constant temperature, RHi and qv would behave the same, i.e. gradients in qv translate directly into gradients of RHi. Nevertheless, the temperature change with height plays an important role for producing sharp gradients (as can be seen in our first case in figure 1). Thus, there is added value for using RHi instead of just qv for the gradient method, also because RHi is a linear variable thus leading to gradients of the same order of magnitude at different temperature regimes. This would not be the case for absolute humidity, which changes over several orders of magnitudes with decreasing temperature.

Moreover, in the manuscript, the term "thermal tropopause" and "lapse rate tropopause" (where the classic WMO definition uses the term lapse rate) mean the same thing. This could be explicitly explained in the manuscript.

We added following text to clarify this:

*This definition is frequently used for studies of the extratropical UTLS and is often referred to as the lapse rate tropopause. In the following, we refer to this tropopause definition as a thermal tropopause.*

Literature

Overall, the impression of the reviewer is that the coverage of the existing literature on the tropopause is better in the previous paper by some of the authors (Köhler et al., 2024, K24 hereafter), than in the present manuscript. For example the classic study by Hoinka (1997) is discussed in K24 but not here. The same is true for the study by Maddox and Mullendore (2018), who point out that there are potentially two different interpretations of the classic WMO lapse rate definition. Further, using potential vorticity (PV) as an entity to be considered when deducing a tropopause is criticized by the argument that choosing a particular PV value introduces a certain arbitrariness; however, when the maximum gradient of PV is considered, this is not the case (Kunz et al., 2011; Turhal et al., 2024).

Both reviewers give us different opinions on the comprehensiveness of the review of previous literature, both of which we can understand. We have therefore tried to make the review of the tropopause a little more precise and focused on mid-latitudes.
In order not to make the introduction too large, we also provide references to overview papers.

Transport barrier for certain compounds The characteristics of the tropopause as a transport barrier might be different for different compounds (e.g., Pan et al., 2018; Bauchinger et al., 2025). From the manuscript it seems that here the focus is on a transport barrier for water vapor (e.g. l. 67). If this is correct, this fact should be obvious in the abstract (and perhaps the title). If this is not correct the manuscript should demonstrate in how far the new tropopause definition is relevant for other trace species (say N2O or CH4).
We agree. We realize that our previous statements about the RHi-GT tropopause acting as a transport barrier were too strong. In this paper, we focus primarily on documenting the occurrence and characteristics of the RHi-GT tropopause, rather than on its role in water-vapor transport. We have clarified this in the abstract and introduction to make the observational focus clear. The relevance of the RHi-GT tropopause for other trace species (e.g., $N_2O$ or $CH_4$) is not addressed here but will be an interesting topic for future studies.

Minor points
l. 6: "approaches": only a comparison with the thermal tropopause is presented here
We have rewritten this to:

*Based on high-resolution radiosonde data, we can show that our RHi-GT-based definition is generally consistent with, and often provides a clearer characterization than, the thermal tropopause*

l. 13: Looks like the entire study is on the extra-tropics
To clarify, we added *mid-latitudes* here.

l. 17: Note that STE (in particularly of H2O) is often only poorly represented in models (Charlesworth et al., 2023).
We added following text:

> *However, it should be noted that there are still uncertainties in the description of exchange processes in large-scale models (Charlesworth et al., 2023; Hoffmann and Spang, 2022).*

Further studies of the vertical gradient of potential temperature as a measure of the tropopause (Kohma and Sato, 2019).
We added the reference. Thank you.

l. 36: I do not agree. Already the old WMO lapse rate definition considers two tropopauses. It should be clear here if the classic double tropopause issue is meant here (e.g., Randel et al., 2007; Castanheira and Gimeno, 2011) or something else.
You are right. We corrected this and rewrote this sentence:

> *While these definitions were originally formulated with synoptic-scale flows in mind, assuming pronounced vertical gradients, they explicitly allow for the identification of multiple tropopauses within a single profile, for example in regions influenced by the jet stream (Pan et al., 2004; Randel et al., 2007; Castanheira and Gimeno, 2011).*

Similar to Köhler et al (2024) we use the WMO criterion without the extension to search for double tropopauses. We added this to the part in Sec 2.2.2:

> *We only search for the first tropopause and therefore use the identical algorithm as in Köhler et al (2024).*

l. 40: The PV values attributed to the location of the tropopause are to some extent arbitrary, I agree. However, this is not the case for more recent studies (e.g. Kunz et al., 2011; Turhal et al., 2024), where the maximum gradient of PV is considered.
We added following sentence:

> *An advancement based on this is the PV gradient method, which avoids the selection of an artificial threshold value (Kunz et al., 2011; Turhal et al., 2024).*

l. 47 ff.: This paper states "In the UTLS we find on average synoptic scale motion, i.e. vertical updrafts in the upper troposphere and subsidence in the stratosphere. These vertical motions induce either adiabatic expansion hence cooling of the air (in case of the tropospheric motion), or even adiabatic compression thus warming of air (in case of stratospheric motion, i.e. the Brewer Dobson circulation. . . " – I am confused here. Clearly the Brewer Dobson circulation is associated with ascending motion in the tropics. This seems in contrast to what is stated here. – Please clarify.
We rewrote this part to:

> *In the UTLS, synoptic-scale vertical motions are relatively weak, but larger-scale mean circulations such as the Brewer–Dobson circulation play a central role: they are*

*characterized by rising air in the tropics, poleward transport in the stratosphere, and descending motion in the mid- and high latitudes (Butchart 2014; Seviour et al. 2011). This results in adiabatic cooling in the tropical ascent region and adiabatic warming at mid- to high-latitudes.*

l. 49: ice particles are also important for short wave radiation.
See next remark for l.50:

l. 50: it is not clear here why the moist layer should be close to the thermal tropopause (where thermal tropopause is the lapse tropopause if I read the paper correctly)
We rewrote this part to:

*Water vapor (and also solid water particles, i.e. ice crystals) is absorbing and re-emitting infrared radiation as an almost ideal black body, leading to a local cooling on top of moist layers situated close to the thermal (i.e. lapse-rate) tropopause. At this level, strong vertical gradients in temperature and humidity make moist layers particularly effective in modifying the local radiative balance (Fusina and Spichtinger, 2010). In addition, ice particles play an important role for the interaction with shortwave radiation, further contributing to the radiative impact of tropopause-near moist layers.*

Actually, the evaluation in Spichtinger et al. (2003) shows that ice supersaturated layer (thus very moist layers) mostly occur close to the (thermal) tropopause.

l. 59: citations for "former investigations"?
We rewrote this part to:

*However, previous studies have shown that diabatic processes, particularly radiative processes, play a crucial role in shaping the tropopause structure (e.g., Randel et al., 2007; Spichtinger, 2014). For instance, the so-called tropopause inversion layer (TIL, Birner et al. (2002, 2006)), a characteristic feature of the UTLS region, known to arise from the interaction of various processes, with radiative cooling by water vapour in the tropopause region making an important contribution and being regarded as the primary mechanism in certain regions or under specific conditions (z.B. Randel et al., 2007; Köhler et al., 2024). Additionally, latent heating from phase changes may induce cirrus cloud convection, thereby altering the vertical structure of the UTLS (Spichtinger, 2014). In a recent study (Köhler et al., 2024), we demonstrated that relative humidity over ice (RHi) is the most suitable quantity for identifying TIL formation, since it inherently accounts for both adiabatic and diabatic processes. Other diabatic processes, such as small-scale turbulence and irreversible mixing, also contribute to the modification, although the exact quantification of their individual contributions to large-scale tropopause structures poses a challenge (e.g. Kunkel et al., 2016). Building on this insight, it is natural to adopt a new perspective and use RHi as the basis for a novel tropopause definition. We propose a concept that defines the tropopause based on RHi gradients, emphasizing its role as a transport barrier for water vapor*

l. 59: Aren't the time scale relevant here? Radiative processes are surely important, but what are the timescales in question here? The location of the tropopause is valid only for a given time span, isn't it?

From former calculations of heating rates (e.g. Fusina & Spichtinger, 2010) we can estimate relevant time scales for the radiation in order of some hours (i.e. O(10h)). Although these scales seem to be quite large, in comparison with the life times of moist layers in the upper troposphere (e.g. life time of ice supersaturation, see Irvine et al., 2014 or Spichtinger et al., 2005) which are of order of few days, the radiation is relevant for shaping the thermodynamic structure. Finally, a recent study (Emig et al., 2025) shows that cirrus clouds in the upper troposphere could alter the tropopause structure, being present for more than 10 hours. Thus, radiation plays definitely a role on its time scales.

l. 60: one could mention Birner's papers (e.g., Birner et al., 2002, 2006) for the TIL;
We added both references (see replay for l.59 about citations.

l. 64: There are a lot of diabatic processes of relevance to the atmosphere so it is not sufficient to include diabatic processes in general in the analysis. Aren't the right processes important here? And this is again a time scale issue.
To keep the introduction concise, we have added this sentence to the section above (l.59 about citations), also regarding Reviewer #2:

> *Other diabatic processes, such as small-scale turbulence and irreversible mixing, also contribute to the modification, although the exact quantification of their individual contributions to large-scale tropopause structures poses a challenge (e.g. Kunkel et al., 2016).*

We agree that there are different time scales to be addressed in a more comprehensive analysis, using explicitly formulated processes (as e.g. in the toy model). However, a concise investigation is beyond the scope of this study, focusing on the observational aspects.

l. 71: vertical gradient
You are absolutely right. We added „vertical"

l. 81: What is the advantage of choosing a regular grid?
A standardised grid simplifies the comparability of several data sets and makes statistical analysis easier. In our present case, this would not have been necessary, but we have already applied this method in Köhler et al. (2024) and want to compare our RHi-GT approach with other data sets in further studies. Therefore, we already use the uniform grid here.

l. 84: "directly measured" – how good is the measurement at low temperature and low water vapor? Can this be quantified? Citations?
We added this information from the manufacturer to the manuscript.

> *For the RS92-SGP, the temperature sensor has a response time < 2.5 s with an uncertainty of 0.5 °C, while the humidity sensor responds within 0.5–20 s with an*

*uncertainty of 5 % RH. In the RS41-SGP, the temperature sensor is faster (≈ 0.5 s) and more accurate (0.2 ◦C), and the humidity sensor responds within 0.3–10 s with an uncertainty of 5 % RH. The RS92-SGP measurement uncertainties are specified by the manufacturer (https://www.bodc.ac.uk/data/documents/nodb/pdf/RS92SGP-Datasheet-B210358EN-F-LOW.pdf, last access: 26 August 2025). The RS41-SGP measurement uncertainties are specified by the manufacturer (https://docs.vaisala.com/v/u/B211444EN-J/en-US,last access: 26 August 2025).*

**Figure 1:** Difference of the mean $N^2$ profile calculated with the Sonntag formula (left) and the Magnus formula (right).

[Figure]

l. 85: citation for the Magnus formula? Is there an assessment of the quality of this formula?

We added the reference of Alduchov and Eskridge (1996). They state that „the errors discussed in this paper are much less than observational errors in humidity values due to the hygrometers…".

Based on your comment, however, we also performed the calculation using the Sonntag (1990) formulation as in Köhler et al. (2024) but did not see any significant changes. For example, the identical level for the RHi-GT is always found in the case studies. Only for the mean $N^2$ profile does the use of Sonntag's parameterisation result in a slightly higher value for the $N^2$ maximum (7.879 $10^{-4}$ s$^{-2}$ compared to 7.809 $10^{-4}$ s$^{-2}$ with the version in the manuscript.)

l. 86. Is there an assessment of the quality of the Murphy and Koop formula?

There is a long discussion of potential uncertainties in the original paper, and we would like to refer to this article, rather than discussing these errors in our study. Nevertheless, we can state that from a visual investigation of their figure 3, the relative error of the saturation vapor pressure over ice is probably less than 5%.

Sec. 2.2: Would not the hygropause also be a valid/important comparison?

That's a good suggestion that we have also thought about. However, in this first study we wanted to compare our version of the tropopause with a version that is often used in the mid-latitudes. Therefore, we limit ourselves here only to the thermal tropopause.

Furthermore, the hygropause is mostly used in the tropics, which is outside the geographical region we want to consider here.

l. 94: threshold for water vapor

We have rewritten this to:

*On the one hand, this could be achieved by applying a simple threshold, for example setting RHi to 10 %*

l. 99: how is this threshold chosen?

This threshold was determined empirically and provided robust results for this radiosonde station. To address Reviewer #2, we additionally examined the sensitivity of our results to the choice of this threshold (see the remarks of Reviewer #2 for line 100)

We added following text:

*Additionally, it is important to note that the choice of the threshold may differ for other geographic regions. Further studies will have to show the influence of the threshold in other regions, such as tropical or polar regions.*

l. 104: a small gradient in RHi means a small gradient in water vapor correct? How can this happen if the troposphere is humid and the stratosphere is dry? Is this an indication that the transport barrier is not very pronounced under these circumstances?

Since relative humidity over ice is temperature-dependent, a small gradient in RHi does not necessarily correspond to a similarly small gradient in specific humidity, but of course it is possible, since RHi depends linearly on qv.

As an illustrative example, the thermal tropopause in Figure 2 is higher than the RHi-GT, since the gradient threshold is met in the lower troposphere. However, there is a significant gradient in the specific humidity, which does not translate as strong in the RHi profile. At the radiosonde station considered here, the profile was influenced by a cut-off low which affects the vertical structure through dynamic processes. Cut-off lows are associated also with stratosphere-troposphere exchange and therefore an erosion of the transport barrier (Price and Vaughan, 1993).

Adjusting the RHi-GT threshold to lower values (-0.05%/m) can shift its detected height and bring it close to the thermal tropopause in this case.

Therefore, further studies should systematically evaluate the RHi-GT approach under varying synoptic conditions to assess its robustness and applicability. However, this goes beyond the scope of this paper

[Figure]

**Figure 2:** Example of a radiosonde measurement under the influence of a cut-off low at 22 October 2016 at 12 UTC. In the lower part a 500hPa geopotential map at the same time is presented showing the cut-off low over Germany. The approximate location of the the radiosonde station is marked with a red star.

.

l. 107: report 42 also in percent.

We added „0.56%"

l. 111: Only 18 "no tropopause cases" vs. 42 (l. 107); report 18 also in percent. Is 18 "better" than 42?
We added „0.24%"
It is without evaluation, only for reasons of transparency

Figures 1-3: How are these cases selected?
We replaced the first sentence of the section to explain the idea:

> *To evaluate the performance of the new tropopause definition, we first examine three representative cases that are intended to cover different relative positions of the RHi-GT with respect to the thermal tropopause, namely situations where the RHi-GT is located above, approximately at, or below the thermal tropopause.*

Fig. 1, caption: state what is temperature (red line I guess) and what is dew point in panel b. The sonde is labeled 12 UTC but the measurements were taken over a certain time period. Report the period.
We added the legend for the different curves and added following text in Sec 2.1

> *Radiosondes typically ascend with a vertical velocity of about 5 m s$^{-1}$ ($\approx$ 300 m min$^{-1}$), such that they reach 16 km altitude within roughly 45–55 minutes (World Meteorological Organization, 2025). The time indicated marks the launch of the radiosonde.*

l. 121: why "definitions" – you only compare with the thermal tropopause
We corrected that.

l. 121: "caveats": this is not completely obvious, the figures indicate that the results are different, not which tropopause is better. Also note that the cold point and the lapse rate tropopause are often not co-located as well.
You are right. We omitted this statement.

l. 122: shown – where?
We added *(Fig. 4)* to the text.

l. 130: which panel shows the "absolute humidity profile"?
We replaced „absolute" with *„specific"* to match the naming of the figure.

l. 137: "the θ " does not sound right
We corrected that.

l. 142; "optical"?
We corrected this and replaced it by „visual"

l. 146: The implicit assumption seems here that the upward transport of water vapor (and the freeze out) occurs along the vertical profile. Of course, this is an over-simplification. To

understand the water vapor at 10 km in (say) Fig. 2 one needs to consider the Lagrangian history of the air parcel (e.g., Schiller et al., 2009; Khaykin et al., 2022), not only what happens below.

Following both reviewers' comments and subsequent discussions with colleagues, we acknowledge that the use of the term transport barrier may be too strong in this context. As correctly noted, a comprehensive assessment would also require an analysis of the air mass history. We therefore refrain from making this conclusion in the revised manuscript. We therefore omitted the second part of the sentence concerning the transport barrier.

ls. 147-148: this "confidence" depends very much on the question how the cases in Figs. 1-3 are selected.

We looked at least into 100 cases, therefore we rewrote this part to:

*These cases, together with a substantially larger number of additional profiles that were analyzed, make us confident to use RHi-GT as a new definition of the extra-tropical tropopause in terms of a transport barrier for water vapor, preferred over the conventional thermal tropopause.*

Fig. 4: I like this statistical comparison in this Figure. Nonetheless, the reference tropopause is only the thermal tropopause here as throughout the paper.

In this manuscript, we focus on determining the tropopause based on the relative humidity gradient and initially compare it only with the thermal tropopause. We agree that further studies should extend this analysis to additional geographical regions and incorporate a broader range of observational and modeling datasets to enable a more systematic evaluation with regard to other tropopause definitions.

l. 155: "prolonged warming" is unclear; explain and/or provide references.

You are right, this was not described very well. We reformulated it to:

*The deviation towards negative height differences, i.e. RHi-GT height is larger than the thermal tropopause, is limited by the dry stratosphere, as in this layer the relative humidity over ice falls below the threshold required for the RHi-GT definition. Therefore, the RHi-GT is not expected to be much higher than the thermal tropopause.*

l. 169: which Figures are these colors referring to?

We do not indicate the colours, as they were also partially incorrect. We apologise for this.

l. 170: results employing RHi for the tropopause determination

We have rewritten this part to clarify that the RHi profiles are investigated here:

*Figure 5 presents the mean RHi profiles relative to the two tropopause definitions. The left panel shows the results with respect to the RHi gradient tropopause (RHi-GT). First, by design, we clearly see a distinct vertical gradient in RHi that captures the abrupt shift between the moist troposphere and the dry stratosphere.*

l. 177: Would Krämer et al. (2020) also be a suitable citation here?

You are right! We added this reference.

 What means "close" here? Approximate value in km?

We replaced „close" with

> *[…] are often found just below the tropopause, within about 0.5–1 km […]*

 why is the kink nonphysical? In individual ascents (Figs. 1-3) there
seem to be also kinks

You are right. We omitted „nonphysical" and added a remark pointing to the fact that the
difference is due to sampling (bold text):

> *While the overall picture is similar to the RHi-GT evaluation, significant differences*
> **due to the different coordinate system, i.e. sampling,** *to the RHi-GT coordinate system
> are obvious. First, the transition from troposphere to stratosphere is less pronounced
> with a broader transition layer. The mean profile shows also a distinctive kink in the
> RHi curve for the thermal tropopause, while the transition in the RHi-GT case is
> somewhat smoother.*

 "signals" in which quantity? Also, does the radiation care about the tropopause? And
means relative to a tropopause height?

We added „radiative heating"

> *[…] strong vertical gradients lead to the strongest signals in radiative heating (see
> e.g., Fusina and Spichtinger, 2010).*

Radiation does not care about the tropopause itself, but responds to the given temperature and
humidity profiles. The strongest vertical gradients of humidity are typically located near the
tropopause and therefore maximize the radiative effect.
Yes, the means are shown relative to the corresponding tropopause height.

 "lower altitude" – should this be visible in Fig. 4? "lower altitude" seems not obvious
from Fig. 4. Suggest some discussion.

We added more information to make the point clearer.

> *First, the mean profile is shifted toward higher temperatures, with the mean RHi-
> GT temperature about 5 K higher than that of the thermal tropopause. While RHi-GT
> generally occurs at similar heights as the thermal tropopause (3,580 cases, see Fig. 4),
> more cases are found below (2,496) than above (1,382) the thermal  tropopause. Since
> temperatures increase with decreasing altitude, RHi-GT tropopauses located below the
> thermal tropopause naturally exhibit higher temperatures, explaining the observed mean
> temperature increase.*

 can you quantify "just above" (in m)?

We replaced „just above" to „*50 m above*".

 "definitions": here only a comparison on the thermal tropopause is presented

Also due to comments by Reviewer #2, we have rewritten this part to:

*Overall, we see that the separation between tropospheric and stratospheric $N^2$ values is better represented by the new RHi-GT definition than by the thermal tropopause definition, which constitutes an additional benefit for studies of the tropopause region. At the same time, interesting differences between the profiles remain, which merit further investigation beyond the scope of this study.*

For the response to the following comment, we want to state again that we completely removed the section with the toy model. Nevertheless, we will response to the comments

l. 241: I cannot find Ra in eq. 3
the text was not correct, here R_v should have been explained.

l, 249: mention cp
ok

l. 257, eq. (7): The time derivative of RHi (using eq. 3) can be expressed . . . (suggest being a bit more explicit here)
yes, we agree

l. 261: note that the updraft is not necessarily vertical.
We slightly disagree, since updrafts are vertically by definition. Nevertheless, we see the point of addressing 3D motions.

l. 264: Yes, clouds are usually located below the tropopause, but not always. For example short term cooling could be induced by upward propagating gravity waves. Could this aspect be of relevance here?
From our investigations, most clouds (or even supersaturated layers) are located below the (thermal) tropopause, see e.g. Spichtinger et al. (2003). However, there are rare cases of ice particles in the stratosphere (e.g. Müller et al., 2015). Short term cooling by gravity waves is certainly an important process for local cooling and thus acting as a trigger for ice nucleation (see, e.g., Spichtinger & Krämer, 2013). Detailed investigations might be carried out in future studies (including the toy model).

l. 265: Is this true also for the situation within a convective cloud, where a tropopause is hard to find (Maddox and Mullendore, 2018).
Good point, this is not clear and subject of future investigations. In the upper part of convective clouds we probably could assume a "moist adiabatic" lapse rate (but with respect to ice, see Spichtinger, 2014)  due to relaxation to saturation by phase changes (i.e. clouds), thus sharp gradients in water vapor might be smeared out.

l. 267: which "layer"?

The layer with enhanced RHi values.

 I think it would be easier to follow these considerations if it is better explained why the specific latent heat L is relevant here if phase changes are neglected.
At this stage, the latent heat just appeared because of the use of the Clausius-Clapeyron equation for expressing the derivative of RHi with respect to temperature in a meaningful way. This is not connected to actual latent heat release.

 Eq 9: I am not sure here: Is this equation true in general or is it formulated neglecting cloud processes?
This equation is true in general, possible cloud processes are present in the change of water vapor, i.e. in the third term (involving dqv/dz)

 why is the second term small?
Because the influence of pressure on the change in relative humidity itself is quite small. Changes in p do not much affect Rhi.

 "all" is not applicable here – the comparison in the arguments presented here is comparsion to the thermal tropopause. For example, if an ozone tropopause is considered (Hoinka, 1997), the thermal structure of the tropopause is not relevant.
We agree

 "diabatic processes" are put forward here; I think the paper could make a stronger point if it states which processes are relevant here. First, I think that friction is neglected. Second, I think that radiative time scales are too slow (do the authors agree?). That leaves phase change (cloud formation) as the central process. If the authors agree, these arguments could be more explicit.
We think, that several diabatic processes on different scales might play a role. We concentrated more on the radiation and cloud processes, sicne they are more accessible, but mixing etc. is also an important candidate. We plan to investigate the relative contributions (also on different scales) of these processes in future.

 change "the apply"
ok

 What are the typical radiative time scales in question here?
Probably order of hours

 How is w determined? Citation?
For synoptic scale motions (e.g. along warm fronts) we can assume vertical velocities of order ~1-5 cm/s … however, a detailed statistics is lacking and should be carried out for a more robust investigation.

For the toy model, also the assumption of a temperature profile is necessary, if I understand correctly – how is it determined?

We used a very idealized profile for a first guess.

l. 289 and 291: "showed" is not correct

ok

Fig. 8, left: the temperature decrease seems to be throughout the troposphere (as far as visible in the figure), more than 1 K per hour. How realistic are such cooling rates?

Synoptic scale motion (e.g. along warm fronts) might last for hours, thus an adiabatic cooling of order ~1K is quite realistic.

l. 306: This analysis assumed the absence of cloud formation .. this should be explicitly stated.

ok

l. 312: but phase changes are neglected in Section 4 as far as I can see. I think the paper could be more explicit about the relative importance of radiation and phase changes regarding how a tropopause location can be determined.

We agree and will use all these suggestions for a separate study, using the toy model again.

l. 330: journal title should not be in capitals

We corrected this.

l. 334: why is "atmosphere" in capitals

We corrected this.

l. 342: there should be no space in the formula of $H_2O$

We corrected this.

Reviewer #2

This paper suggests that relative humidity, or more specifically, the vertical gradient of relative humidity, be used as an alternative definition of the tropopause. Actually this seems to me to be an interesting and potentially useful idea and I have not heard this definition of the tropopause suggested before. So I am certainly in favour of a paper on this topic being published. However I feel that the current version of the paper has some substantial flaws on which I give comments below. Therefore I recommend that the paper is published only after major revision that addresses my comments (or refutes them in a convincing way).

The major points where I think that the paper could be improved (and it would be a pity if it was not) are the following:

Tropopause vs Extratropical Tropopause:
I believe that this new definition applies to the extratropical tropopause. The authors do not show any examples applying it to the tropical tropopause and my guess is that it would not work so well. So I recommend that 'Tropopause' is replaced by 'Extratropical Tropopause' in the title. It certainly isn't necessary to add 'extratropical' to every mention of 'tropopause' in the paper -- but its occasional use would be helpful.
We agree. Our study focuses on the mid-latitude tropopause, and we have updated the title accordingly. We also made it more clear throughout the manuscript that we focus in this first description of the RHi-GT on the mid-latitudes.

Transport barriers:
The abstract states (and there are similar statements at various places in the main text): 'The tropopause acts as a transport barrier between troposphere and stratosphere ...' -- well yes and no. My own view is that the tropopause is should be regarded as air mass boundary -- in particular air below the tropopause (tropospheric air) has experienced a very different recent history to air above the tropopause (stratospheric air). This pattern of transport may arise because of the presence of transport barriers, but these need not coincide with the tropopause it self. For example, one view of the extratropical tropopause (the part poleward of the subtropical jet) is that it results from the fact that on lower isentropes there is free exchange with the subtropics whereas on higher isentropes there is not free exchange because the subtropical jet acts as a barrier to latitudinal transport. The barrier to transport (the jet) and the part of the extratropical tropopause being considered are not co-located. Of course the term 'tropopause as transport barrier' is used in reputable review articles and it would not be appropriate for this referee report to become a kind of independent review article on the nature of the tropopause. But one concrete reason for caution in using this term is that it seems to me to encourage a 1-dimensional view of the extratropical tropopause, with this being an upper limit to strong vertical transport in the troposphere. Most would agree that this is not a satisfactory model for the extratropical tropopause, not least since the relevant transport in the troposphere is not transport in the vertical.
The fact is that what you are proposing is potentially useful definition of the tropopause whether or not one defines it as a transport barrier, or as an air mass boundary, or as any

variation on those. So I recommend dropping the very strong emphasis on tropopause as transport barrier.

Following the reviewer's detailed comment, as well as feedback from the other reviewer and discussions with colleagues, we agree that calling the tropopause a transport barrier was too strong. As the reviewer points out, the tropopause is better seen as an air-mass boundary, and transport barriers such as the subtropical jet do not necessarily line up with the tropopause itself. The simple 1-dimensional view of the extratropical tropopause as the top limit of vertical transport is clearly too simplistic. We have therefore removed statements referring to the tropopause as a transport barrier and focus on documenting the occurrence and characteristics of the RHi-GT tropopause. The relevance of transport processes associated with the RHi-GT will be addressed in future studies. Importantly, the RHi-GT definition remains potentially useful regardless of whether the tropopause is interpreted as a transport barrier, an air-mass boundary, or any other concept.

General comments on the tropopause and related topics:
Some of the general, e.g. review, comments on the tropopause and its structure seem a bit muddled to me. Of course there is room for uncertainty in this general area -- e.g. views of which processes are most important vary across scientists working in this area. But I think that the paper would be more effective if it said less, so that the reader can focus on the interesting idea of using RHi (or its vertical gradient) to identify the tropopause, rather than continually thinking 'my view of how this works is different to that expressed here'.

Both reviewers give us different opinions on the comprehensiveness of the review of previous literature, both of which we can understand. We have therefore tried to make the review of the tropopause a little more precise and focused on mid-latitudes.
In order not to make the introduction too large, we also provide references to overview papers.

Relation to other tropopause identifiers:
To me the primary explanation for the RHi structure is that in the upper troposphere the water vapour concentrations are primarily determined by relatively local temperature conditions, whereas in the stratosphere they are primarily determined by remote temperature conditions. (The latter was the key insight of the landmark paper by Brewer.) The difference in transport pathways that leads to this difference in RHi is similarly responsible for differences in concentrations of species such as ozone and carbon monooxide. Therefore the RHi approach offers the possibility of identifying this chemical or air mass tropopause using information available from standard radiosonde measurements, rather than requiring in-situ chemical measurements.
I am surprised that the authors don't emphasise this point more. (But perhaps they don't agree with me on this point.)

We thank the reviewer for this insightful comment and agree that the RHi structure in the upper troposphere mainly reflects local temperature conditions, while stratospheric water vapor is more influenced by remote conditions, as noted by Brewer. We also recognize that these differences in transport pathways affect other species like ozone and carbon monoxide, and that the RHi approach could help identify a chemical or air-mass tropopause using standard

radiosonde data. In this paper, however, we focus on documenting the RHi-GT tropopause itself, and a more detailed look at the underlying processes will be left for a follow-up study.

Theoretical model:
The current description of this model is not suitable for publication. (See further comments below.)

DETAILED COMMENTS:

Abstract: 'While common definitions rely on quantities conserved under adiabatic changes, diabatic effects ... are also decisive for the tropopause structure' -- this seems to muddle several different points. If we take the use of PV, for example, the advantage of PV (conserved on moderate timescales) over temperature gradient (not conserved on moderate timescales) is that it provides an identification of the tropopause that does change on short time scales through purely reversible dynamics. (Another way of saying this would be that according to the temperature definition an air parcel might be in the troposphere one day, the stratosphere the next, and the troposphere next, and this was recognised many years ago as not consistent with other characteristics of troposphere versus stratosphere.

Actually, PV is based on dry air, thus phase changes are not taken into account. There is still a debate if the concept of PV can be extended to include moist processes. Recent work by Kooloth et al. (2022, 2023) show that moist processes can be included but then conservation laws looks different than for the dry case. This, for instance, shows how difficult it is to use the concepts from dry dynamics in extension to diabatic processes. Using the relative humidity leads to a less complicated procedure.

L24: 'most popular' -- 'most used'? Of course the reason why it is the most used definition is that it was proposed by the WMO and is therefore a natural standard definition for many purposes -- e.g. for researchers who are not particular interested in the tropopause itself, but in some other aspect of the atmosphere which requires a distinction between troposphere and stratosphere.
We changed the sentence to:
> *One of the most widely used definitions is the so-called thermal tropopause […]*

L26: 'Another definition ...' -- this jumps to the cold point definition but it is essentially irrelevant to the discussion here because it is a tropical definition and, as I have noted above, the emphasis of your paper is on the extratropics.
You are right. We omitted this sentence and noted later:
> *The reader is referred to Köhler et al. (2024), Bauchinger et al. (2025), and Hoinka (1997) for detailed reviews of different* tropopause definitions, including some that are not applied in the present study because they are primarily relevant for tropical regions, such as the cold-point tropopause.

L35: This might be a good point to cite Tinney et al MWR 2022 who specifically discuss the fact that high-vertical resolution causes problems for this definition.

Thank you for this comment. We added this sentence:

> *It should be noted, however, that very high vertical resolution can pose challenges for this approach, as discussed by Tinney et al. (2022).*

L36: 'In reality, several tropopauses can [sometimes] be found in one profile with this type of definition ...' -- but we know this makes physical sense (e.g. filaments of tropospheric air from the subtropics overlying stratospheric air). If you are suggesting here that a useful definition of the tropopause is REQUIRED to identify only one tropopause in a given column then I would say that I do not accept that -- some kind of 'operational' specification of the tropopause should make it clear that sometimes identifying multiple tropopauses is the only sensible outcome.

Thank you for pointing that out - as also Reviewer #1. We rewrote this part to:

> *While these definitions were originally formulated with synoptic-scale flows in mind, assuming pronounced vertical gradients, they explicitly allow for the identification of multiple tropopauses within a single profile, for example in regions influenced by the jet stream (Pan et al., 2004; Randel et al., 2007; Castanheira and Gimeno, 2011).*

L47: This very brief description of the dynamical processes contributing to tropopause structure/maintenance is very focused on vertical motion -- upwelling in troposphere/downwelling in stratosphere -- which is a drastic oversimplification (in my view) -- e.g. see reviews such as Gettelman et al (2011), or papers such as Kunkel et al (2016 ACP) which emphasise the three-dimensionality of the dynamics affecting the tropopause. I think that these two or three sentences could straightforwardly be changed to be consistent with that.

Reviewer #1 had a similar comment and we rewrote this part to:

> *IIn the UTLS, synoptic-scale vertical motions are relatively weak, but larger-scale mean circulations such as the Brewer–Dobson circulation play a central role: they are characterized by rising air in the tropics, poleward transport in the stratosphere, and descending motion in the mid- and high latitudes (Butchart 2014; Seviour et al. 2011). This results in adiabatic cooling in the tropical ascent region and adiabatic warming at mid- to high-latitudes.*

L61: 'is assumed to be crucially created by radiative cooling ...' -- I don't see this as a universally held view. The Randel et al (2007) paper focused on high latitudes. Other papers have shown that models can produce a TIL without inclusion of the specific radiative properties of water vapour -- e.g. Son and Polvani (2007 GRL) -- but there are probably more recent papers of this type -- and again see Kunkel et al (2016) who don't specifically emphasise the radiative role of water vapour over other processes.

We agree and reworked this section:

> *However, previous studies have shown that diabatic processes, particularly radiative processes, play a crucial role in shaping the tropopause structure (e.g., Randel et al., 2007; Fusina and Spichtinger, 2010; Spichtinger, 2014). For instance, the so-called tropopause inversion layer (TIL, Birner et al. (2002, 2006)), a characteristic feature of the*

*UTLS region, known to arise from the interaction of various processes, with radiative cooling by water vapour in the tropopause region making an important contribution and being regarded as the primary mechanism in certain regions or under specific conditions (z.B. Randel et al., 2007; Köhler et al., 2024). Additionally, latent heating from phase changes may induce cirrus cloud convection, thereby altering the vertical structure of the UTLS (Spichtinger, 2014). In a recent study (Köhler et al., 2024), we demonstrated that relative humidity over ice (RHi) is the most suitable quantity for identifying TIL formation, since it inherently accounts for both adiabatic and diabatic processes. Other diabatic processes, such as small-scale turbulence and irreversible mixing, also contribute to the modification, although the exact quantification of their individual contributions to large-scale tropopause structures poses a challenge (e.g. Kunkel et al., 2016). Building on this insight, it is natural to adopt a new perspective and use RHi as the basis for a novel tropopause definition. We propose a concept that defines the tropopause based on RHi gradients, emphasizing its role as a transport barrier for water vapor*

L97: It seems to be taken as obvious here that standard radiosondes provide water vapour measurements in the tropopause region that are sufficiently accurate to be useful. I had the impression in the past that standard radiosondes were not felt to be sufficiently accurate to provide useful stratospheric water vapour measurements -- but perhaps that applies more to the tropics and to concentrations of 10ppmv or less? Does the fact that it is RHi rather than concentration that is being used help with this potential problem -- if it exists?

It is correct that the standard radiosondes have issues with low concentrations of water vapor at low temperatures, especially for dry conditions in the stratosphere. However, there is rather a dry bias than a moist bias. In addition, the main structure of the vertical profile, i.e. the gradients can be seen clearly in the RS41 radiosonde data. Since we investigate relative humidity instead of absolute humidity, and rather the vertical gradient than the absolute values of RHi, our method is less sensitive to the dry bias of the radiosonde. This will probably also be an advantage when we investigate vertical profiles from other data sets, as e.g. ERA5 data, which are known to have a dry bias, too.

L100: 'Vertical Relative Humidity Gradient' would be more precise (and an important aspect of the paper is your emphasis on RELATIVE humidity).

You are right, we added „Relative".

L100: My guess is that the identification of the tropopause is insensitive to the precise choice here. Presumably you tested that? Please confirm. What values would be too high or too low? The same goes for the choice of relative humidity threshold. Please give more explicit information. Having looked at the examples given in Figures 1-3, I am wondering if a criterion of RHi > 20% without the gradient criterion would work as well as your chosen criterion. Did you try possibilities of this type?

We want to highlight that the idea of our manuscript is to present the observation that the vertical gradient of RHi is suitable as a tool for determining the midlatitude tropopause. Since

we use only data from one geographical location, the presented values are presumably not the most favorable everywhere.

The choice of parameters should of course be analysed in more detail in subsequent publications. Special focus should also be placed on vertical profiles from model data (e.g. ERA5). Other parameters are to be expected there due to the coarser model resolution.

As we have omitted the previous Section 4 on modeling (see general statement), we have instead included a section on the sensitivity of the threshold values, which addresses your questions.

In the first attempts for this manuscript we also used simple fixed values for the determination of the tropopause height, including a value of RHi=20% as suggested by you. We did not pursue this further as we focused on the feature of the strong RHi gradient. However, we also repeated the analysis with the simplified determination of a pure fixed value based on your question and obtained surprisingly good agreement with our results for this 10-year data set (see left panel of Fig. 3).

Figure 3: Same as for Fig. 7 in manuscript but left panel determined only with a threshold value of RHi=20% without gradient criterion.

[Figure]

However, when defining the tropopause height using a simple threshold based on RHi, it is important to account for measurement accuracy, as probes with different precisions may yield different results. In contrast, identifying the tropopause based on a pronounced RHi gradient is less sensitive to such measurement variations.

L106: For clarity it would be best if this 'break' criterion was highlighted in the same way as the other two criteria.
We added this a third point in the list:

> *Tropopause Height Assignment: As soon as both criteria are met, the corresponding height is stored as the tropopause height RHi-GT.*

L109: 'We use the same code as ...' -- at first sight this seems unnecessary. But I guess the key point is the requirement for smoothing mentioned a couple of lines later. These two sentences could be combined.
We combined both sentences:

> *Note that the high resolution data of temperature raise issues in terms of determining the lapse rate; a smoothing of the profiles is applied for extracting the large scale feature of thermal profiles. For this we use the identical algorithm as in Köhler et al. (2024).*

Figure 1: In the 2nd panel temperature and dew point are presumably red/blue respectively.
Thank you for pointing that out. We corrected the images in the manuscript accordingly.
As an example, here the new Fig. 1.

[Figure]

L127: For reasons given above I don't find it helpful to use 'transport barrier' here -- why not say 'the RHi-GT'.
We changed it to „RHi-GT".

L137: 'the theta' -- delete 'the'
We corrected this.

L139: My guess is that these characteristics -- i.e. relatively low static stability in air that is stratospheric from an airmass perspective -- are typical of certain synoptic situations. For

example the air in the 8-12 km range could have relatively high PV if cyclonic vorticity was compensating for the low static stability. (But certainly in this example the RHi characterisation appears to capture a clear 'air mass' tropopause.)

This is an interesting remark, we will investigate this in future investigations dedicated to more specific cases.

L141: replace 'optical' by ,visual'.

We corrected this.

L143: 'The WMO thermal tropopause therefore appears somewhat arbitrary at this point.' -- one could say the same about the RHi tropopause. There are multiple relatively moist shallow layers -- presumably filaments penetrating from the subtropics. The RHi approach chooses one of these. Whether this is the best choice, or a better choice than the WMO thermal tropopause, is not very clear. But certainly the RHi approach is no worse than the thermal tropopause.

We agree that, as with the WMO thermal tropopause, any practical definition of the RHi tropopause involves a certain degree of arbitrariness. In our case, the choice of threshold is indeed somewhat conventional. However, it is grounded in the physical characteristic of a transition from a very dry layer above to a substantially more humid layer below. By focusing on the strongest gradient from above, the RHi-based definition captures the fundamental contrast that separates two "spheres": a dry stratosphere and a moister troposphere. In this sense, the RHi tropopause is not less well defined than the thermal tropopause, but rather highlights the moisture-driven transition that complements the temperature-based perspective.

L155: I don't understand 'limited by the dry stratosphere, as the WMO criterion is based on prolonged warming'.

You are right, this was not described very well. We reformulated it to:

> *The deviation towards negative height differences, i.e. RHi-GT height is larger than the thermal tropopause, is limited by the dry stratosphere, as in this layer the relative humidity over ice falls below the threshold required for the RHi-GT definition. Therefore, the RHi-GT is not expected to be much higher than the thermal tropopause.*

L181: '(and nonphysical)' -- I don't see the justification for 'nonphysical' -- the fact is that both panels are based on the same RHi data -- the difference between them results from the fact that the data has been organised in different ways. There is nothing 'nonphysical' about that. Of course it might be that one panel is more straightforward to explain or provides a simpler conceptual model -- but that is not 'physicality' vs 'non physicality'.

You are right. We omitted „nonphysical" and added a remark pointing to the fact that the difference is due to sampling (bold text):

> *While the overall picture is similar to the RHi-GT evaluation, significant differences* **due to the different coordinate system, i.e. sampling,** *to the RHi-GT coordinate system are obvious. First, the transition from troposphere to stratosphere is less pronounced with a broader transition layer. The mean profile shows also a distinctive kink in the RHi curve for the thermal tropopause, while the transition in the RHi-GT case is somewhat smoother.*

L219: 'more unsteady pattern' -- poorly chosen wording -- there is nothing steady or unsteady about this pattern.

You are right, we replaced this with

*[…] a structurally more complex pattern […]*

L228: 'Thus, we can finally state that a much clearer separation of tropospheric and stratospheric N squared values has been achieved by simply using ...' -- I don't see the justification for 'much clearer'. But certainly there are very interesting differences between the two profiles and further work (beyond the scope of this paper) is needed to understand them.

We agree that this statement is too strong, so we have replaced it with this (bold text):

*[…] we can finally state that **a more distinct** separation of tropospheric […]*

Additionally, at the end of the section we added that further studies are needed to understand the differences.

*Overall, we see that the separation between tropospheric and stratospheric $N^2$ values is better represented by the new RHi-GT definition than by the thermal tropopause definition, which constitutes an additional benefit for studies of the tropopause region. At the same time, interesting differences between the profiles remain, which merit further investigation beyond the scope of this study.*

Section 4:

I found this section quite difficult to follow and have made general rather than line-by-line remarks.

You seem to be considering a model in which quantities vary in height (z) and time. Time derivatives seem to be material derivatives -- is that correct? You neglect 'cloud processes' meaning that RHi is a function of T and p, for given $q_v$, leading to (9). Then there is a jump to vertical gradients rather than rate of change of time in (10) when you conclude that it is vertical gradients of temperature and $q_v$ that primarily determine the vertical gradient of RHi. (I don't think that many would be surprised by that.)

The text L278-284 -- including 'huge benefit of our new definition', 'For definitions it is even worse' -- seems unjustified. Many of the statement are muddled. For example, using PV as a basis for tropopause definition does not 'neglect' diabatic processes. It is a com-bination of diabatic and dynamical processes that set the overall structure of the PV field and make it suitable as a basis for identifying the tropopause. Then the fact that PV is ma-terially conserved on short timescales is advantageous because it means (as noted above) that a given air parcel does not suddenly change from being tropospheric to stratospheric (or vice versa) through purely reversible dynamics (which is possible according to the lapse-rate approach). Also, to correct a simple factual error -- the (vertical) gradient of potential temperature is NOT, as you state, conserved under adiabatic processes.

We agree that we expressed some of the statements about PV in a sloppy way. Nevertheless, we think that PV is a valid concept for dry air but the extension to moist processes is difficult; there is still a debate how to extend PV to a moist version, and recent work by Kooloth et al. (2022, 2023) shows that even with a consistent extension by phase transitions, the conservation

properties change completely. As stated above, we delete the whole original section 4 from the manuscript, including a sensitivity analysis instead.

The toy model is mysterious. You are imposing a positive w in the UT, and a 'typical radiative cooling contribution at the upper part of the humid layer'. No details of the latter are given, but it is clearly playing a crucial role, since, as far as I can tell, it is the sole mechanism for giving any temperature change about 10km. I don't believe that q_v is changing in time (true?), so the whole response in relatively humidity (and its vertical gradient) has to be understood in terms of the temperature dependence of the saturation vapour pressure. You refer to the 'temperature' and 'humidity' terms but because the humidity is kept constant the 'humidity' term changes only through the change in temperature. The main physical effect being demonstrated seems to be that because UT temperatures are decreasing the UT RHi is increasing and therefore the vertical gradient of RHi becomes more negative.
I don't see what this model calculation is supposed to demonstrate. Is it that a strong negative vertical gradient in RHi is somehow self-reinforcing? But the physical ingredients included in the model -- essentially imposed cooling in the troposphere -- don't see a plausible starting point.
My overall comment on Section 4 is that the motivation for the model needs to be much clearer -- what is the hypothesis motivating this model? -- and the details needed to be stated more clearly, so that someone else could reproduce the results for themselves if they wished.
We agree that section 4 was not adequately motivated and the toy model was described too vaguely. Since this study has a more descriptive and observational character, we decided to leave the toy model out of the study. We will investigate the processes in more details in a later study with a better treatment of the toy model.

**References**

Alduchov, O. A. and Eskridge, R. E.: Improved Magnus Form Approximation of Saturation Vapor Pressure, J. Appl. Meteor., 35, 601–609,350
https://doi.org/10.1175/1520-0450(1996)035<0601:IMFAOS>2.0.CO;2, 1996

Sonntag, D.: Important new values of the physical constants of 1986, vapour pressure formulations based on the ITS90, and psychrometer formulae, Z. Meteorol., 40, 340–344, https://doi.org/10.1127/metz/3/1994/51, 1990.

Price, J. D. And Vaughan, G.: The potential for stratosphere-troposphere exchange in cut-off-low systems, Quart J Royal Meteoro Soc,
https://rmets.onlinelibrary.wiley.com/doi/10.1002/qj.49711951007, 1993

Irvine, E. A., B. J. Hoskins, and K. P. Shine (2014), A Lagrangian analysis of ice-supersaturated air over the North Atlantic, J. Geophys. Res. Atmos., 119, 90–100, doi:10.1002/2013JD020251

Spichtinger, P., K. Gierens, H. Wernli, 2005: A case study on the formation and evolution of ice supersaturation in the vicinity of a warm conveyor belt's outflow region. Atmos. Chem. Phys., 5, 973-987.

Nicolas Emig, Annette K. Miltenberger, Peter M. Hoor, and Andreas Petzold: Impact of cirrus on extratropical tropopause structure. EGUsphere, https://doi.org/10.5194/egusphere-2024-3919, 2025

Kooloth P, Smith L M, Stechmann S N, 2023: Hamilton's principle with phase changes and conservation principles for moist potential vorticity. *Q. J. Roy. Met. Soc.* **149**, 1056-1072.

Kooloth P, Smith L M, Stechmann S N, 2022: Conservation laws for potential vorticity in a salty ocean or cloudy atmosphere. *Geophys. Res. Lett.* **49**, e2022GL100009.

---

## Referee Report (RR1)

**Re-review of "The Frosty Frontier: Redefining the mid-latitude Tropopause using the Relative Humidity over Ice"**

BY PHILIPP REUTTER AND PETER SPICHTINGER

**General**

As stated in the first review, this paper reports a novel way of deducing a tropopause using the relative humidity over ice (RHi), which is good. The paper discusses the tropopause in Northern hemisphere mid-latitudes, which is now reflected in the title (and throughout the paper). Overall, the paper has been revised thoroughly.

I also note that focus of the paper is less on the tropopause as a transport barrier (e.g. for  $N_2O$  or  $CH_4$ ) and that the relation of the new tropopause and the thermal (lapse rate) tropopause have been clarified.

I think the authors have adequately replied to the questions and suggestions of reviewer 1 (and also reviewer 2). I suggest that the paper is accepted now (any very small remaining changes can be done in a revised submission).

**Minor points**

- comment on l. 40: I think the word "artificial", which is introduced here is too strong; perhaps "picking a particular value of PV" or similar
- comment on 1. 59 (former investigations); avoid "z.B." in the text of the manuscript
- comment on l. 59 (radiation): I am not arguing against radiation here, but some of these arguments could be helpful in the paper as well.

---

## Author Response (AR2)

**Reply to Reviewer**

We would like to thank both reviewers once again for taking the time to review the revised version of the manuscript. We are grateful for their suggestions, which have led to a significantly improved manuscript!

In the following, we answer to the comments point by point. Questions and remarks of the reviewers are marked in orange, reply of the authors are marked in black and changes to the manuscript are marked in blue.

**Reviewer #1**

**General**

As stated in the first review, this paper reports a novel way of deducing a tropopause using the relative humidity over ice (RHi), which is good. The paper discusses the tropopause in Northern hemisphere mid-latitudes, which is now reflected in the title (and throughout the paper). Overall, the paper has been revised thoroughly. I also note that focus of the paper is less on the tropopause as a transport barrier (e.g. for N2O or CH4) and that the relation of the new tropopause and the thermal (lapse rate) tropopause have been clarified.

I think the authors have adequately replied to the questions and suggestions of reviewer 1 (and also reviewer 2). I suggest that the paper is accepted now (any very small remaining changes can be done in a revised submission).

**Minor points**

- comment on I. 40: I think the word "artificial", which is introduced here is too strong, perhaps "picking a particular value of PV" or similar

We have changed it as suggested.

comment on I. 59 (former investigations); avoid "z.B." in the text of the manuscript

"z.B." was only in the document to the reviewers, it was already corrected in the manuscript.

- comment on I. 59 (radiation): I am not arguing against radiation here, but some of these arguments could be helpful in the paper as well.

**We added following text based on our reply:**

Radiative heating rate estimates (Fusina and Spichtinger, 2010) suggest characteristic time scales of several hours O(10 h), which are relevant compared to the lifetimes of upper-tropospheric moist layers (e.g. Irvine et al., 2014; Spichtinger et al., 2005). Recent results (Emig et al., 2025) also indicate that long-lived cirrus clouds can modify the tropopause structure, confirming the key role of radiation on these time scales.

**Reviewer #2**

In revising this paper the authors have resolved many of the concerns I expressed in my first review (and my impression is that they will similarly have resolved many of the concerns of the other referee). I repeat my previously expressed view that this new definition of the tropopause is an interesting and potentially useful idea and given the revisions I recommend that this paper is published without undue delay.

I do have some comments on this version of the paper that I recommend are considered by the authors before the paper proceeds to publication.

My main critical comment is that whilst in their reply to my original comment on the use of the term 'transport barrier' the authors state:

'Following the reviewer's detailed comment, as well as feedback from the other reviewer and

discussions with colleagues, we agree that calling the tropopause a transport barrier was too strong. ... We have therefore removed statements referring to the tropopause as a transport barrier and focus on documenting the occurrence and characteristics of the RHi-GT tropopause. ...'

But in fact there are some sentences where the term 'transport barrier' remains and is used in pretty much the same way as before. For the reasons given in my previous reviews I think that the paper would be much better if these sentences were removed.

We thank the reviewer for pointing this out. A few instances of "transport barrier" were inadvertently retained. We will review the manuscript and remove these occurences to ensure full consitenc with the revised discussion.

More generally, and while the paper has improved significantly in revision, I still think that the paper would benefit from the 'less is more' principle -- if something does not have to be said, then it is better not to say it. Many of my detailed comments are along these lines.

**DETAILED COMMENTS:**

L50: 'oxid' to 'oxide'
We corrected this.

L53-55: Ordering might be better as: 'including some, such as the cold-point tropopause, that are not applied in the present study because they are primarily relevant for tropical regions' We have changed it as suggested.

L63: 'close to the thermal (i.e. lapse-rate) tropopause' -- I am wondering why this statement is not simply 'close to the tropopause'? We have changed it as suggested.

L64-69: There seems to be some redundancy between these two sentences -- before and after the paragraph break -- and a paragraph break at this point does not seem necessary.

Together with the comments from Reviewer 1, we expanded the previous paragraph slightly with regard to radiation and then added the statement about "friction" at the end, so that the short paragraph is now omitted.

On the other hand, diabatic processes play a role. Water vapor (and also solid water particles, i.e. ice crystals) is absorbing and re-emitting infrared radiation as an almost ideal black body, leading to a local cooling on top of moist layers situated close to the tropopause. At this level, strong vertical gradients in temperature and humidity make moist layers particularly effective in modifying the local radiative balance (Fusina and Spichtinger, 2010). In addition, ice particles play an important role for the interaction with shortwave radiation, further contributing to the radiative impact of tropopause-near moist layers. Radiative heating rate estimates (Fusina and Spichtinger, 2010) suggest characteristic time scales of several hours O(10 h), which are relevant compared to the lifetimes of upper-tropospheric moist layers (e.g. Irvine et al., 2014; Spichtinger et al., 2005). Recent results (Emig et al., 2025) also indicate that long-lived cirrus clouds can modify the tropopause structure, confirming the key role of radiation on these time scales. Finally, friction and irreversible mixing, as e.g. driven by turbulence, contribute to the change in variables as diabatic processes.

L71: I don't really see why the thermal lapse-rate tropopause definition 'relies exclusively on adiabatic processes'.

We have changed it to "mostly".

L64-85: Looking at these paragraphs, there seems to be a certain amount of repetition -- e.g. radiative effects mentioned in L65 and then again in L77-79. My general feeling is that the authors are trying to claim too much for the justification of their new tropopause definition -- from my point of view the new definition is an effective way of identifying the tropopause as an air mass boundary -- considering how this helps understand the processes that maintain and determine the structure of the tropopause comes later.

We have summarized and shortened this section

Yet diabatic processes - including radiative cooling, latent heating, and small-scale mixing - also shape the tropopause structures, as exemplified by the tropopause inversion layer (TIL) (e.g., Birner et al., 2002; Randel et al., 2007; Fusina and Spichtinger, 2010; Spichtinger, 2014; Köhler et al., 2024; Kunkel et al., 2016). Relative humidity over ice (RHi) captures both adiabatic and diabatic contributions to TIL formation, motivating a tropopause definition based on RHi gradients.

L85-86: 'emphasising its role as a transport barrier for water vapour' -- but, as noted above, you have previously agreed in your response that the use of 'transport barrier' is too strong and have removed statements that the tropopause is a transport barrier'.

We omitted this statement.

L80: ... Kohler et al (2024) demonstrated ...?

This part has been rewritten.

L129: trivial point, but this derivative must of course be negative

You are right, we corrected this.

The derivative of RHi with height ( $\partial$ RHi/ $\partial$ z) must be smaller than -0.15% m-1.

L135: List the break criterion as an extra numbered criterion?

We added the break criterion as the fourth point and omitted the following paragraph.

4. Break Criterion: To avoid unrealistically low tropopause heights, any detected height below 3000 m is discarded and no value is assigned for that profile.

Note, the RHi-GT algorithm has problems with profiles that show only very small gradients in RHi. It can happen that the criterion for the gradient only switches on very far down in the profile and delivers unrealistic values. Therefore, a break criterion has been introduced to avoid unrealistically deep tropopause heights lower than 3000 m. In such a case, no value is specified for the tropopause.

L138: 'Further studies will have to show the influence of the threshold in other regions such as tropical or polar regions.' But I thought that we (two reviewers + authors) were all agreed that the tropical tropopause is very different to the mid-latitude tropopause, so finding a useful criterion for the tropical tropopause is not simply going to be a case of 'changing the threshold'.

You are absolutely right and we therefore omitted this sentence.

L174: 'In summary, this indicates a strong transport barrier ...' -- contradicts your claim to have remove mention of transport barriers.

We rewrote this to:

In summary, this indicates a strong **confinement** of water vapor at the RHi-GT level, whereas the conventional thermal tropopause definition fails completely to detect this feature.

Figures 2 and 3: in looking at these cases I am again wondering about the synoptic situations that give rise to these situations where the RHi-GT is well above or well below the thermal tropopause. But you have said that this will be the subject of future investigations.

Yes, the examples are intended to illustrate different ways in which RHi-GT and TTP behave.

L181: \*\*\* 'transport barrier' \*\*\*!

We rewrote this to:

These cases, together with a substantially larger number of additional profiles that were analyzed, make us confident to use RHi-GT as a new definition of the extra-tropical tropopause in terms of **water vapor distribution**, preferred over the conventional thermal tropopause.

Figure 5: Probably not very surprising -- but nonetheless reassuring -- that an RHi-based tropopause signal gives a clearer signal in RHi.

Yes.

L255: \*\*\* 'transport barrier' \*\*\*!

We deleted this.

L256-266: This paragraph summarises the differences in the N^2 profile as seen according the two tropopause definitions and then concludes that a 'more distinct separation ... has been achieved by simply using the gradient of relative humidity ...' Is that really the case? It is not what

I see when I simply look at the two panels of Figure 7. This seems to be an example of the authors feeling that they have to claim that the RHi-GT definition is 'better'. I think that the differences between the two distributions shown in the two panels of Figure 7 prompt all sorts of interesting questions about the nature of the TIL and the structure of the TIL that appears in profiles constructed from the thermal tropopause definition, but on the basis of the information presented I cannot see clear evidence that a more distinct separation of tropospheric and stratospheric N^2 values has been achieved using the RHi-GT definition.

You are right, we reformulated this part to be more neutral:

The original definition of the TIL using the thermal tropopause shows a structurally more complex pattern directly below the thermal tropopause. While the tropospheric values of the Brunt-Väisälä frequency are similar for both tropopause definitions and thus normalized profiles, a noticeable kink is visible at the local minimum of  $N^2 \approx 2 \cdot 10^{-4} \text{s}^{-2}$ . This kink can be seen in many studies regarding the TIL (Birner et al., 2002; Gettelman et al., 2011; Köhler et al., 2024). Starting from this kink,  $N^2$  rises sharply and then reaches its maximum 50 m above the thermal tropopause with  $N^2 \approx 7 \cdot 10^{-4} \text{s}^{-2}$ . In contrast, for the RHi-GT definition, the  $N^2$  maximum coincides with the tropopause level itself. Above this level,  $N^2$  decreases more gradually for the thermal tropopause definition than for RHi-GT, leading to similar values only about about 3000, m higher.

These differences suggest that the RHi-GT and thermal tropopause definitions emphasize different structural aspects of the TIL. While the RHi-GT-based profiles appear smoother and more vertically confined, the thermal tropopause definition highlights a more extended transition region.

Section 4: I think that the information presented here is a useful addition to the paper. Of course what one might hope would be that there would be a range of values of the RHi threshold and the vertical gradient value for which the corresponding tropopause level would be relatively insensitive. Figure 9 seems to imply quite a bit of difference for the three choices of gradient shown -- perhaps not surprising for the 0.01% criterion but the difference between 0.15% and 0.25% more surprising. Figure 10 right panel is more encouraging -- for the choice of 0.15% the altitude seems very insensitive to the min (RHi) criterion. (This is just a comment -- no response needed.)

We thank the reviewer for the careful inspection of the sensitivity analysis. We agree that the differences between the 0.15 % and 0.25 % gradient thresholds appear somewhat pronounced. This is mainly due to the steepness of the RHi gradients in the tropopause region, where small changes in the threshold can lead to noticeable shifts in the detected level. Nevertheless, as also noted by the reviewer, the overall sensitivity remains limited for reasonable choices of the gradient threshold (around 0.15 %), and the definition is quite robust against variations in the minimum RHi criterion.

L297: 'threshold for this particular dataset' -- of course what one hopes is that the a threshold can be selected that works for many extratropical locations -- the new criterion would be much less useful if it had to be 'tuned' for different locations.

We fully agree with the reviewer that a broadly applicable threshold is essential for the usefulness of the RHi-GT definition. In this study, we indeed optimized the threshold based on the available dataset to illustrate the concept, but the goal is to identify a criterion that remains robust across different extratropical regions in the future.

L306: As already commented, it is not very clear to me that the RHi-GT criterion shows a 'much sharper and clearer transition between troposphere and stratosphere' with regard to N^2.

We reformulated this part to:

When analyzing thermodynamic variables over 10 years relative to the RHi-GT tropopause height, RHi and static stability ( $N^2$ ) show a more coherent transition between tropospheric and stratospheric conditions than when referenced to the thermal tropopause.